# Study of Asian indexes by a newly derived dynamic model

**Tsung-Jui Chiang-Lin**[1], **Yong-Shiuan Lee**[2]*, **Tzong-Hann Shieh**[3], **Chien-Chang Yen**[4], **Shang-Yueh Tsai**[5]

**1** Department of Finance, Minghsin University of Science and Technology, Hsinchu, Taiwan, **2** Department of Statistics, National Chengchi University, Taipei, Taiwan, **3** Department of Aerospace and Systems Engineering, Feng Chia University, Taichung, Taiwan, **4** Department of Mathematics, Fu-Jen Catholic University, New Taipei City, Taiwan, **5** Graduate Institute of Applied Physics, National Chengchi University, Taipei, Taiwan

* 99354501@nccu.edu.tw

**Data Availability Statement:** The data underlying the results presented in the study are available from Refinitiv Datastream. The Refinitiv Datastream is purchased by National Chengchi University, and all faculty and students of NCCU have access to this database. Those who are

## Abstract

We take the stock prices as a dynamic system and characterize its movements by a newly derived dynamic model, called the new Price Reversion Model (nPRM), for which the solution is derived and carefully analyzed under different circumstances. We also develop a procedure of applying the nPRM to real daily closing prices of a stock index. This proposed procedure brings a different perspective to the study of stock prices based on thermodynamics, and the time varying coefficients in the nPRM offer economic meanings of the stock movements. More specifically, the average of smoothed historical data $A$ in the nPRM, analogous to the environment temperature in the Newton's law of cooling, represent an implied equilibrium price. The heat transfer coefficient $\kappa$ is adapted to be either negative or positive, which illustrates the speed of convergence or divergence of stock prices, respectively. The empirical study of ten Asian stock indexes shows that the nPRM accurately characterizes and forecasts the market values.

## Introduction

The stock index, a key variable reflecting the economic status in an area, is a popular research topic in macroeconomics. The market prices of an individual stock reflect the company's existing value or future profit growth. A stock index, measured by the weighted mean prices of selected stocks in a stock market, represents the stock market's performance and potential competitive ability. Investors who have different motives for participating in the market concern about the changes of a stock or a stock index's market prices of different scales, intraday high-frequency data, daily closing prices, weekly or monthly summary, and etc. Accurately forecasting the stock movements could help the investors with decisions about investment strategy.

Whether the stock prices can be forecasted has been repeatedly discussed in finance [1, 2]. The efficient market hypothesis [3] forms a foundation of the theory that under what circumstances the stock prices is predictable. The efficient market hypothesis states that three types of

interested in getting access to the data can purchase or request for the product from https://www.refinitiv.com/en or https://reurl.cc/xEg67E.

**Funding:** The authors received no specific funding for this work.

markets can be distinguished as the weak form, the semi-strong form, and the strong form depending on the level of market efficiency. The stock prices fully reflect all available market information in a strong efficient market, thus they are not predictable. The martingale depicts the stock movements in the strong efficient markets by the theory of random walks [4]. In contrast, many real markets are considered as the weak or the semi-strong efficient markets [2, 5]. Consequently, to forecast the prices in these markets is possible.

The analysis of the stock movements is based on either the stock prices or the stock returns. The choice between the stock prices and the returns mainly depends on the approach adopted. From the financial accounting perspective, fundamental analysis [6, 7] and technical analysis [8] are the two main intellectual traditions. The fundamental analysis evaluates the stocks by measuring their intrinsic values from overall economic and industrial conditions. The fundamental analysis assumes that the stock prices are linearly correlated to the predictors in the financial statement. As a result, the linear regression models are usually applied to forecast the stock prices. On the contrary, the technical analysis assumes that all known fundamentals are factored into prices. Hence, charting past fluctuations in the stock indexes allows the analyst to identify the patterns and trends. Technical analysis forecasts the market movements through popular techniques like chart analysis, cycle analysis and computerized technical trading systems [8].

Various statistical methods are prevalent in the history of financial academic researches. In econometric modeling, applying the autoregressive integrated moving average (ARIMA) models and vector autoregression (VAR) models is common. When using ARIMA and VAR models, the stationarity of the time series is required. Therefore, the stock returns instead of the stock prices are the primary research subject [9–11]. Financial ratios like the log dividend-price ratio [12, 13], the dividends [9], the price–earnings ratio [10, 13], the dividend yield [10, 14], the consumption-wealth ratio [15], the book-to-market ratio [10] serve as important predictors for asset returns in regression models, particularly VAR models. Although some good predictors in financial literature have restricted predictive ability for the equity premium due to model uncertainty and instability [11], combining individual forecasts produce out-of-sample gains [16]. Modeling stock returns reduces low-frequency information, the trend for example, from the perspective of signal processing. Therefore, forecasting the stock prices based on models constructed from detrended data may be less accurate in this sense.

The stock prices can be treated as signals. From the signal processing perspective, the forecasts can be constructed by decomposing the signals into features and putting them into artificial intelligence models. The artificial intelligence models applied to the stock prices in existing studies include support vector regression (SVR) [17], neural networks [18–22], tree-based models [23], etc. To applying the artificial intelligence models, acquiring features is an important step. The empirical mode decomposition (EMD) [17, 21], the ensemble empirical mode decomposition (EEMD) [24], and the complementary ensemble empirical mode decomposition (CEEMD) [22] are widely employed methods to decompose the stock prices, which are regarded as signals. The empirical results of US and UK stock indexes show that the artificial intelligence models are better than the ARIMA and GARCH models in terms of forecasting directional symmetry [21]. Furthermore, the CEEMD-PCA-LSTM model [22], which integrate the CEEMD, principal component analysis (PCA), and long short-term memory (LSTM) networks, outperforms naive recurrent neural networks (RNN) or LSTM networks in terms of forecasting accuracy for six selected stock indexes. Still, in a review study of the returns from financial perspective [25], the empirical evidences show that no single model out of statistical, artificial intelligence, frequency domain, and hybrid models under investigation could be applied uniformly to all markets. On the other hand, the direction prediction problem

is another task, which can be tackled by the tree-based models [23] to predict the stock price direction.

This paper considers a physical-mathematical method, the approach of the differential equations and the dynamic systems. From thermodynamics, we adopt the Newton's law of cooling to describe the mean reverting of the stock index prices. In recent years, modeling the stock index from the physical perspective by treating it as a dynamic system is emerging [26–29]. To the best of our knowledge, the first attempt to study the stock index, the Taiwan Stock Exchange Capitalization Weighted Stock Index (TAIEX), as well as the options through dynamic models [26], which are derived based on parabola approximation [30]. Another study [24] also considers the stock markets as complex dynamical systems, but their interactions are analyzed by methods of signal processing, the EMD and the EEMD along with detrended cross-correlation analysis (DCCA). Existing studies recognize the nonlinear behaviors of the stock movements [31–34]. However, the models fits [33, 34] are obtained via statistical method during a period of up trend or down trend.

Based on the related work [27], we consider to apply the Price reversion model (PRM), which has a form of the Newton's law of cooling, to the stock markets. Yet, the empirical evidence in [27] shows that the PRM tends to underestimate the forecasts. Therefore, we identify the reasons of inaccurate forecasts by the PRM and propose to model the stock prices by a newly derived model, i.e. the new Price Reversion Model (nPRM). For the same dynamic model described using the differential equation in the PRM, we derive its solutions rigorously so that the solutions of the nPRM can better fit the realistic stock prices of the markets than the PRM. Two major contributions of the nPRM are as follows. Firstly, the nPRM seeks for the solutions that will not only achieve higher prediction accuracy but also provide useful economic meanings (refer to Figs 1 and 2 below). The coefficients in the nPRM, $A$ and $\kappa$, represent the implied equilibrium price, and the convergence speed to the implied equilibrium price, respectively. Secondly, for many predictive models involved in the time series analysis, it is necessary to transform the stock prices to other economic variables, usually the returns or the price increments. The nPRM can make the predictions directly for the stock prices, which implies that the model considers all daily information in the analysis. Furthermore, the conduction of the proposed procedure is simple and convenient.

## Methods

We treat the stock index, denoted as $S(t)$, as a dynamic system, which is a differentiable function with respect to time $t$. We model the subject by the nPRM, represented by a differential equation as follows.

$$\begin{cases} \dfrac{dS(t)}{dt} & = & f(S(t)) & = & \kappa\left[S(t) - A\right], \\[2mm] \dfrac{d^2S(t)}{dt^2} & = & \kappa\dfrac{dS(t)}{dt}, \\[2mm] S(t_0) & = & S_0, & S(t_1) & = & S_1, \end{cases} \tag{1}$$

where $S(t)$ is the value of the subject at time $t$, $A$ and $\kappa$ are known constants for $t \in [t_0, t_1]$. We further denote $v(t) = \frac{dS(t)}{dt}$ and $a(t) = \frac{d^2S(t)}{dt^2}$, which represent the velocity and acceleration of the dynamic system. The system along with its velocity and acceleration at specified time points $t_0$ and $t_1$ are denoted as $(S(t_0), v(t_0), a(t_0)) = (S_0, v_0, a_0)$ and $(S(t_1), v(t_1), a(t_1)) = (S_1, v_1, a_1)$ respectively for simplicity.

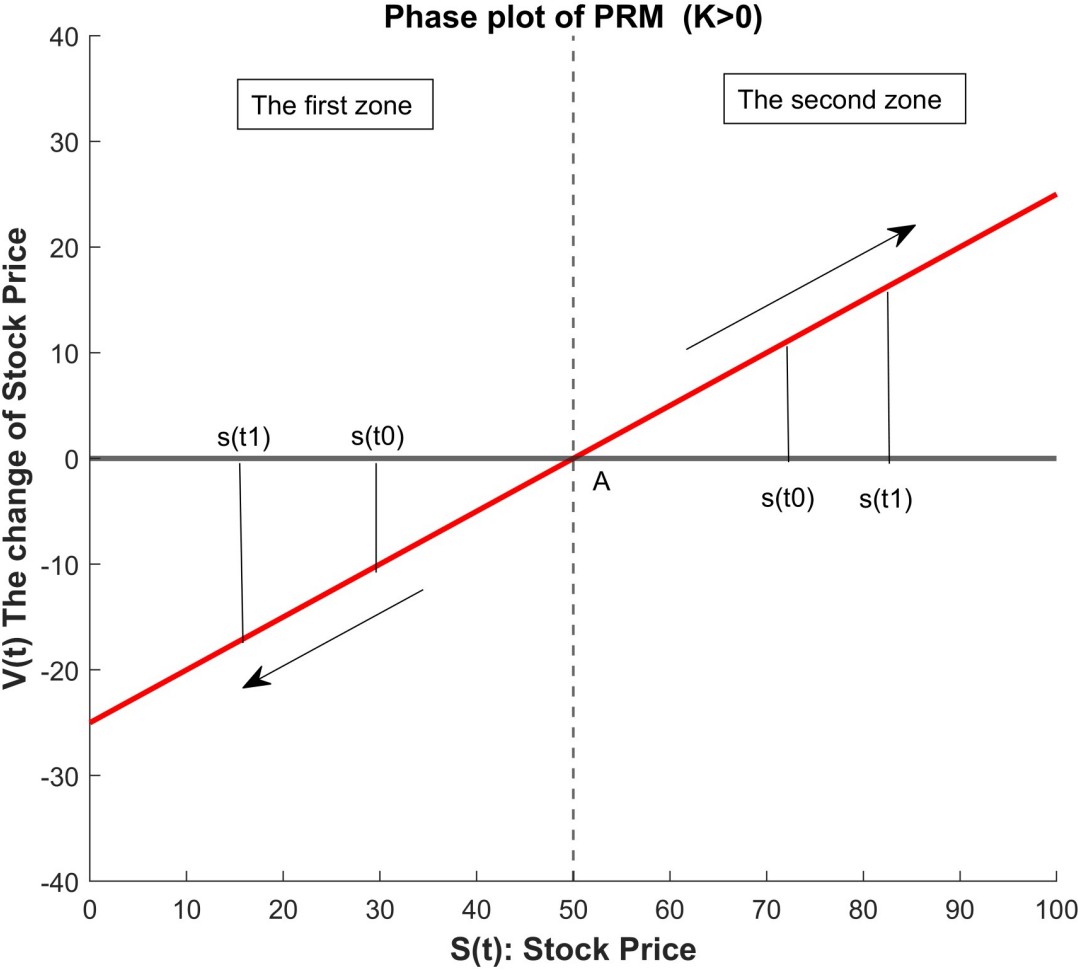

**Fig 1. The phase plot of the nPRM for $\kappa > 0$.**

The Eq (1) is adapted from the Newton's law of cooling [35]. In the Newton's law of cooling, $\kappa$ is always negative so that the temperature of the subject converges to the environment/surrounding temperature. Additionally, when the difference between the subject temperature and the environment temperature is large, the absolute value of $\kappa$ is large and the speed of convergence is high. When the subject temperature is closer to the environment temperature, the absolute value of $\kappa$ is small, which results in a slower convergence.

Consider the differential Eq (1), $\kappa$ and $A$ are constant during the period $t \in [t_0, t_1]$. Generally they are time related in a dynamic system for a longer time period. Furthermore, the time-related coefficients are often very dynamic in real-world data. For applications, the coefficients are assumed to be constant in a relatively small interval with given values between time $t_0$ and $t_1$.

For the consistence of this paper, let us first review the PRM in the following subsection.

## The Price Reversion Model (PRM)

In the existing study [27], the PRM assumes that the external force equals zero at $A$. The second derivative is ignored in the Eq (1). Depending on the values of $S(t)$, $S_0$, and $A$, four cases are considered and the solutions are listed as follows.

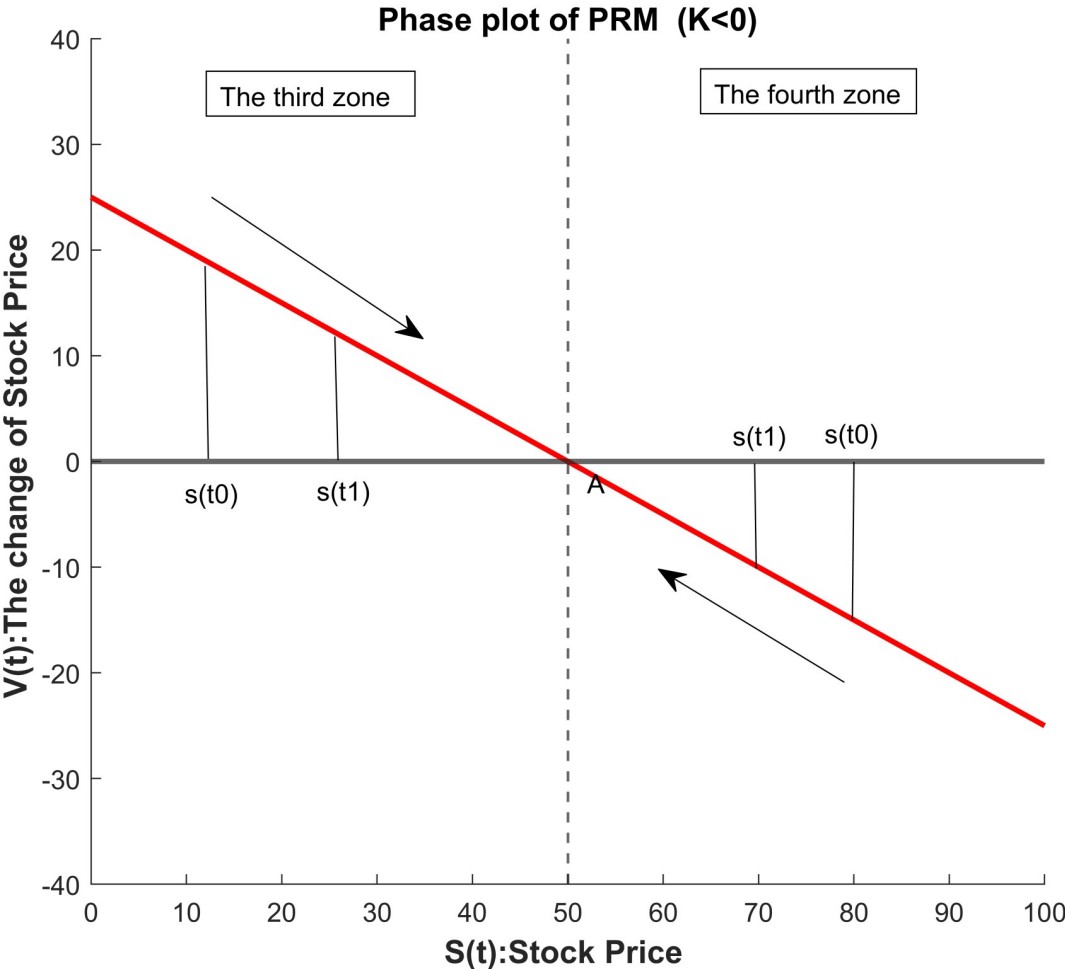

**Fig 2. The phase plot of the nPRM for $\kappa < 0$.**

Case A.
 If $S(t) > S_0$ and $(S(t) - A) \cdot (S_0 - A) > 0$,

$$S(t) = A + (S_0 - A) \cdot e^{\kappa(t - t_0)}. \tag{2}$$

Case B.
 If $S(t) > S_0$ and $(S(t) - A) \cdot (S_0 - A) < 0$,

$$S(t) = A - (S_0 - A) \cdot e^{\kappa(t - t_0)}. \tag{3}$$

Case C.
 If $S(t) < S_0$ and $(S(t) - A) \cdot (S_0 - A) > 0$,

$$S(t) = A + (S_0 - A) \cdot e^{-|\kappa|(t - t_0)}. \tag{4}$$

Case D.

If $S(t) < S_0$ and $(S(t) - A) \cdot (S_0 - A) < 0$,

$$S(t) = A - (S_0 - A) \cdot e^{-|\kappa|(t-t_0)}. \tag{5}$$

However, essentially the situations represented by the Eqs (3) and (5) does not exist since the external force equals zero at $A$. It motivates us to re-derive the nPRM in the next subsection.

## Derivation of solutions to the new Price Reversion Model (nPRM)

In Eq (1), $A$ is an equilibrium price of a stock index for a specific market, which is analogous to the environment temperature, and $\kappa$ is a growth coefficient. The force on the stock price draws it near the equilibrium if $\kappa < 0$ or pushes it away from the equilibrium if $\kappa > 0$. The nPRM is a more generalized dynamic system than that described by the Newton's law of cooling. Therefore, in the nPRM $\kappa$ can be either positive or negative. When $\kappa < 0$, the nPRM is similar to the thermodynamic system; when $\kappa > 0$, the behavior of the dynamic system is a different story.

If $\kappa > 0$ in (1), $A$ is a repeller. Fig 1 is the phase plot of nPRM when $\kappa > 0$. The first zone demonstrates the situation that the force pushes the price $S(t)$ away from $A$ if the initial price $S_0$ is smaller than $A$. In the second zone, the force repels the price $S(t)$ from the equilibrium price $A$ if the initial price is larger than $A$. On the contrary, if $\kappa < 0$ in (1), $A$ is an attractor. Fig 2 is the phase plot of nPRM when $\kappa < 0$. When the initial price is higher than $A$, the force draws the price to the equilibrium price $A$ as the figure demonstrates in the third zone. In the fourth zone, if the initial price is lower than the equilibrium $A$, the price converges to $A$.

We solve the Eq (1) of the nPRM piecewisely as follows. We divide $S(t)$ into intervals by the critical point, $A$, where $\frac{dS(t)}{dt}\big|_{S=A} = 0$. As a result, $S(t)$ is strictly monotonic in each of the intervals. From Eq (1), after the change of variables we have

$$\int_{S(t_0)}^{S(t)} \frac{dr}{(r-A)} = \int_{t_0}^{t} \kappa dr.$$

Therefore,

$$\int_{S_0}^{S(t)} \frac{dr}{(r-A)} = \ln|r-A|\big|_{S_0}^{S(t)} = \ln[S(t) - A] - \ln(S_0 - A) = \kappa(t - t_0).$$

Then, we have

$$S(t) = A + (S_0 - A) \cdot e^{\kappa(t-t_0)}, \quad t \in [t_0, t_1]. \tag{6}$$

Case 1.

Given $S_0 < A$ and $\frac{dS(t)}{dt} < 0$ for $t \in [t_0, t_1]$ as in the first zone in Fig 1, we obtain $\kappa > 0$. Therefore, the theoretical value $S(t)$ is strictly decreasing in $t$, and $S(t) < A$ for $t \in [t_0, t_1]$. The solution blows up since

$$\lim_{t\to\infty} S(t) = \lim_{t\to\infty} [A + (S_0 - A) \cdot e^{\kappa(t-t_0)}] = -\infty.$$

Case 2.

Given $S_0 > A$ and $\frac{dS(t)}{dt} > 0$ for $t \in [t_0, t_1]$ as in the second zone in Fig 1, we obtain $\kappa > 0$. Therefore, the theoretical value $S(t)$ is strictly increasing in $t$, and $S(t) > A$ for $t \in [t_0, t_1]$. The solution blows up since

$$\lim_{t\to\infty} S(t) = \lim_{t\to\infty} \left[ A + (S_0 - A) \cdot e^{\kappa(t-t_0)} \right] = \infty.$$

Case 3.

Given $S_0 < A$ and $\frac{dS(t)}{dt} > 0$ for $t \in [t_0, t_1]$ as in the third zone in Fig 2, we obtain $\kappa < 0$. Therefore, the theoretical value $S(t)$ is strictly increasing in $t$, and $S(t) < A$ for $t \in [t_0, t_1]$. The solution converges to $A$ since

$$\lim_{t\to\infty} S(t) = \lim_{t\to\infty} \left[ A + (S_0 - A) \cdot e^{\kappa(t-t_0)} \right] = A.$$

Case 4.

Given $S_0 > A$ and $\frac{dS(t)}{dt} < 0$ for $t \in [t_0, t_1]$ as in the fourth zone in Fig 2, we obtain $\kappa < 0$. Therefore, the theoretical value $S(t)$ is strictly decreasing in $t$, and $S(t) > A$ for $t \in [t_0, t_1]$. The solution converges to $A$ since

$$\lim_{t\to\infty} S(t) = \lim_{t\to\infty} \left[ A + (S_0 - A) \cdot e^{\kappa(t-t_0)} \right] = A.$$

A comparison of the cases on different conditions between PRM and nPRM is given by Table 1. As mentioned before, cases B and D in the PRM do not exist due to zero external force at $A$. By examining the phase plots of Figs 1 and 2, the case A of the PRM can be further divided into cases 2 and 4 of the nPRM. The case C of the PRM can be further divided into cases 1 and 3 of the nPRM. We derive solutions for the cases 3 and 4 of the nPRM, which can not be obtained identically from the solutions of the PRM. As a result, the theoretical values obtained via the PRM are not so accurate as those from the nPRM.

## Implementation details of the nPRM

The differential Eq (1) characterizes a dynamic system, whose movement is continuous and at least second-order differentiable. However, the real data of daily stock prices consist of two parts, the trajectory of the dynamic system and the noise. The trajectory of the dynamic system

**Table 1. Comparisons between solutions of PRM and nPRM under different conditions.**

| Model | Case | $S_0 - A$ | $S(t) - A$ | $S(t) - S_0$ | $\frac{dS(t)}{dt}$ | $\kappa$ | Solution | Phase plot |
|---|---|---|---|---|---|---|---|---|
| PRM | A | + | + | + | + | + | Eq (2) | second zone in Fig 1 or fourth zone in Fig 2 |
| | B | - | + | + | + | + | Eq (3) | no existence |
| | C | - | - | - | - | + | Eq (4) | first zone in Fig 1 or third zone in Fig 2 |
| | D | - | + | - | - | - | Eq (5) | no existence |
| nPRM | 1 | + | - | - | - | + | Eq (6) | first zone in Fig 1 |
| | 2 | + | + | + | + | + | Eq (6) | second zone in Fig 1 |
| | 3 | - | - | + | + | - | Eq (6) | third zone in Fig 2 |
| | 4 | + | + | - | - | - | Eq (6) | fourth zone in Fig 2 |

is the functional part while the noise is random. Modeling the noisy data, the stock index for instance, by a differential equation is not appropriate. The reasonable approach is to model the functional movement that can be characterized by the differential equation after filtering out the random noise. We establish a procedure of 4 main steps for applying the nPRM to daily stock prices as follows.

**Step 1—Smoothing.** Moving average is a simple and commonly used filtering method, and hence we implement this method on stock data. We filter out the noise of the data, and define the smoothed stock prices as

$$\tilde{S}(t) = \frac{\sum_{i=0}^{M-1} S(t-i)}{M}. \tag{7}$$

Afterwards, we apply model (1) to the smoothed data and obtain the theoretical values. In order to obtain accurate calculation of $\kappa$, the adequate smoothing period $M$ has to be specified. For this purpose, we estimate the smoothing period $M$ in the training set, which we will further explain later.

This smoothing step is not involved in the procedure of PRM [27]. Nevertheless, directly modeling the market prices of a stock index suffers from severe fluctuations in coefficient values resulted from random part of the price movements, the Brownian motion. Characterizing the smoothed stock prices by the differential Eq (1) is more reasonable so that this step is crucial in our proposed method nPRM.

**Step 2—Calculating A.** To solve (1) we have to know the fixed value of $A$ for a given time period $[t_0, t_1]$ in advance. However, in reality the equilibrium price $A$ is unknown so that the estimation of $A$ is required. $A$ is calculated using smoothed historical data before the period $[t_0, t_1]$. More specifically, we calculate $A$ as

$$A = \frac{\sum_{i=0}^{N-1} \tilde{S}(t_0 - i)}{N}, \tag{8}$$

that is the average of stock prices during a period earlier than $t_0$. The length of the time interval $[t_0 - N + 1, t_0]$ should be carefully chosen. We also determine the time period $N$ based on the training set.

**Step 3—Calculating and controlling $\kappa$ value.** During the solving of model (1), the value of $\kappa$ is first calculated from

$$\frac{d\tilde{S}(t)}{dt} = \kappa[\tilde{S}_0 - A], \tag{9}$$

by substituting $\tilde{S}(t)$ for $S(t)$ in (1). The value of $\kappa$ is constant during a short period $[t_0, t_1]$. However, it changes over time, and thus we further denote it as $\kappa_t$, $t = t_0, \cdots, t_\tau$ for different time intervals. When $\kappa_t > 0$ and has large value, the theoretical value at time $t$ will explode. Nevertheless, this situation does not correspond with the changes of stock prices in the real market. To avoid generating unreasonable theoretical values, we restrict the values of $\kappa_t$ lower than an upper bound.

A gradient control method [36] is proposed to control the values of the gradient in a reasonable range. The $\kappa_t$ is a one-dimensional gradient in the nPRM, and we apply this method to $\kappa_t$ and restrict its value as

$$\kappa_t^* = \begin{cases} \kappa_{t,Q3}, & \text{if } \kappa_t > \kappa_{t,Q3}, \\ \kappa_t, & \text{otherwise}, \end{cases} \tag{10}$$

where $\kappa_{t,Q3}$ is the third quantile from previous values of $\kappa_i$, for $i = t - j, \cdots, t - 1$ and $j \geq 100$ in the series we obtained. The pool of $\kappa_i$'s comprises at least 100 values from historical data so that we can obtain stable values of $\kappa_t^*$.

**Step 4—Making $\tau$-step forecasts.** The theoretical values of the stock prices during time $[t_0, t_1]$ are obtained by the solution (6), in which $S_0$ are replaced by $\tilde{S}_0$. Assuming the coefficients unchanged for a longer period $[t_0, t_\tau]$, we can obtain theoretical values before time $t_\tau$. For discrete time representation, we obtain theoretical daily closing prices at $t = t_0, t_1, \cdots, t_\tau$ as the $\tau$-step forecasts.

## Model evaluation

The forecasting error is measured by three deviation-type measurements, the Mean Absolute Percentage Error (MAPE), the Mean Absolute Error (MAE), and the Root Mean Square Error (RMSE), along with three trend-type measurements, directional Symmetry (DS), correct uptrend (CP) and correct down-trend (CD). The definitions are as follows.

$$MAPE = \frac{1}{T}\sum_{i=t_1}^{T}|\frac{\hat{S}(i) - S(i)}{S(i)}| \cdot 100,$$

$$MAE = \frac{1}{T}\sum_{i=t_1}^{T}|\hat{S}(i) - S(i)|,$$

$$RMSE = \sqrt{\frac{1}{T}\sum_{i=t_1}^{T}(\hat{S}(i) - S(i))^2},$$

$$DS = \frac{1}{T}\sum_{i=t_1}^{T}a_i \cdot 100, \quad a_i = \begin{cases} 1, & \text{if } (\hat{S}(i) - \hat{S}(i-1))(S(i) - S(i-1)) \geq 0, \\ 0, & \text{otherwise,} \end{cases}$$

$$CP = \frac{1}{T_1}\sum_{i=t_1}^{T_1}b_i \cdot 100, \quad where$$

$$b_i = \begin{cases} 1, & \text{if } (\hat{S}(i) - \hat{S}(i-1)) > 0 \text{ and } (\hat{S}(i) - \hat{S}(i-1))(S(i) - S(i-1)) \geq 0, \\ 0, & \text{otherwise,} \end{cases}$$

$$CD = \frac{1}{T_2}\sum_{i=t_1}^{T_2}c_i \cdot 100, \quad where$$

$$c_i = \begin{cases} 1, & \text{if } (\hat{S}(i) - \hat{S}(i-1)) < 0 \text{ and } (\hat{S}(i) - \hat{S}(i-1))(S(i) - S(i-1)) \geq 0, \\ 0, & \text{otherwise,} \end{cases}$$

where $\hat{S}(i)$ is the theoretical value of the model at time $i$, $S(i)$ is the market value of the data at time $i$, $T$ is the length of the forecasts, $T_1$ is the number of data points belonging to up trend, and $T_2$ is the number of data points belonging to down trend.

A smaller value of the MAPE indicates that the model better describes the movements of the stock prices. The MAE and the RMSE show similar tendency with the MAPE. DS measures the frequency of correct directional forecasts. CP and CD measure the frequency of correct directional forecasts when real market values rise and fall, respectively. The larger values of DS, CP, and CD suggest that the model obtains more correct directional forecasts in general, for uptrend, and for downtrend, respectively.

## Results

For application to real data, we split the data into a training set and a testing set, consisting of the earlier 80% and the latest 20% of the data, respectively. We use the training set to specify adequate smoothing period and the time interval for estimation of the equilibrium, that is, the appropriate choices of $M$ and $N$ in (7) and (8), respectively.

### Applied data

The empirical study was conducted based on information from real daily closing price data of ten stock indexes from 2009 to 2019. The indexes include Taiwan TAIEX, Japan Nikkei 225, Korea KOSPI, Hong Kong Hang Seng Index, the Philippines PSEi Index, Thailand SET Index, India S&P BSE SENSEX, Singapore Straits Times Index (STI), Indonesia JKSE, and Malaysia KLCI. The data are collected from Refinitiv Datastream. Each stock index is viewed as a dynamic system and characterized by the nPRM separately.

We implemented the nPRM with a few choices of $M$ and $N$ to obtain $\tau$-step forecasts for the training dataset of each stock index. The values of $M$ range from 2 to 10 while the choice of $N$ includes 30, 60, 90, 120, and 180. We set the number of forecasting steps $\tau = 5$. We assess all combinations of $M$ and $N$ by the criterion of the grand mean of MAPEs across 1-step to 5-step forecasts in the training set. Afterwards, we select the combination of the best performance as the coefficient setting for prediction in the testing set. Table 2 shows that the best combination of $M$ and $N$ with the smallest grand mean of MAPEs in the training set for all the ten indexes. In the following, the forecasts are conducted with $M$ and $N$ specified for each stock index in Table 2.

We conduct all the analyses on a computer with an Intel(R) Core(TM) i7–8565U Processor at 1.80GHz with 24.0 GB RAM running Windows 10. All computations are performed using R version 4.0.3 [37] and the R package `Metrics`.

### Empirical results

The main reason of underestimation of the PRM in the existing study [27] is the difficulty in determining the sign of the growth coefficient, $\kappa$, during the integration. In the proposed nPRM procedure, we derive the solution to the differential equation of the nPRM and propose adequate economic explanations of $\kappa$. Additionally, when $\kappa > 0$ we control its value by a gradient control method [36], and the nPRM is therefore more suited to real markets.

**Table 2. The best combination of $M$ and $N$ in the training set.**

|  | $M$ | $N$ |  | $M$ | $N$ |
|---|---|---|---|---|---|
| **Hang Seng** | 4 | 180 | **SET** | 6 | 180 |
| **Nikkei 225** | 5 | 180 | **BSE SENSEX** | 5 | 180 |
| **KOSPI** | 4 | 180 | **STI** | 4 | 180 |
| **TAIEX** | 4 | 180 | **JKSE** | 8 | 180 |
| **PSEi** | 7 | 180 | **KLCI** | 4 | 180 |

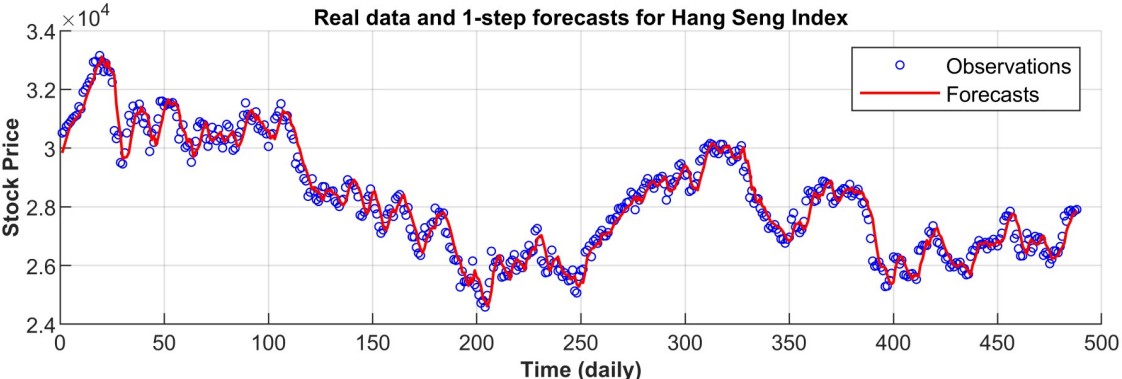

**Fig 3. 1-step forecasts of the nPRM and market values for Hong Kong Hang Seng Index.**

Fig 3 displays the 1-step forecasts of the nPRM for Hong Kong Hang Seng Index along with the market values. Fig 3 suggests that the nPRM possesses accurate forecasting ability. The patterns of the forecasts from 2-step to 5-step by the nPRM are all similar to Fig 3, and thus we omit the demonstration here. Figs 4–12 illustrate the 1-step forecasts of the nPRM for other Asian stock indexes individually, which also indicate accurate forecasts by the nPRM.

Table 3 displays the MAPEs of applying the investigated methods to all of the ten Asian Indexes from 1-step forecasts to 5-step forecasts. Since the performances of three deviation-type errors (MAPE, MAE, and RMSE) are similar, we simply report and discuss the results of the MAPEs. From Table 3, the martingale (see details in S1 Appendix) has the smallest errors followed by those from the nPRM, and the curve fitting (see details in S2 Appendix) has the worst performance.

Table 4 displays the directional symmetry (SD) of the nPRM and the curve fitting from 1-step forecasts to 5-step forecasts for ten Asian stock indexes. The trend-type errors cannot be applied to the forecasts of the martingale since the martingale forecasts the price of the next day by the present price. Therefore, we only compare the nPRM and the curve fitting here. Table 4 shows that nPRM and the curve fitting have similar percentage of correct trend forecasts. The differences between up-trend forecasts and down-trend forecasts of the nPRM and the curve fitting are also not significant (see details in S1 and S2 Tables).

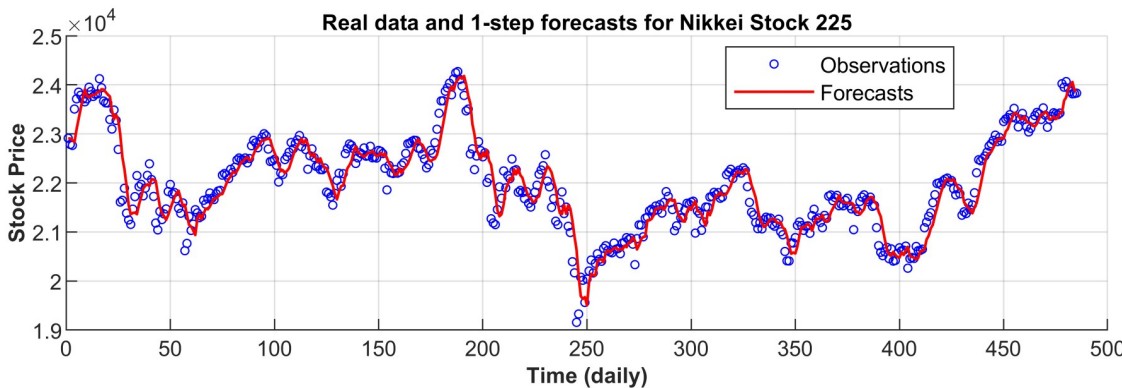

**Fig 4. 1-step forecasts of the nPRM and market values for Japan Nikkei 225 Index.**

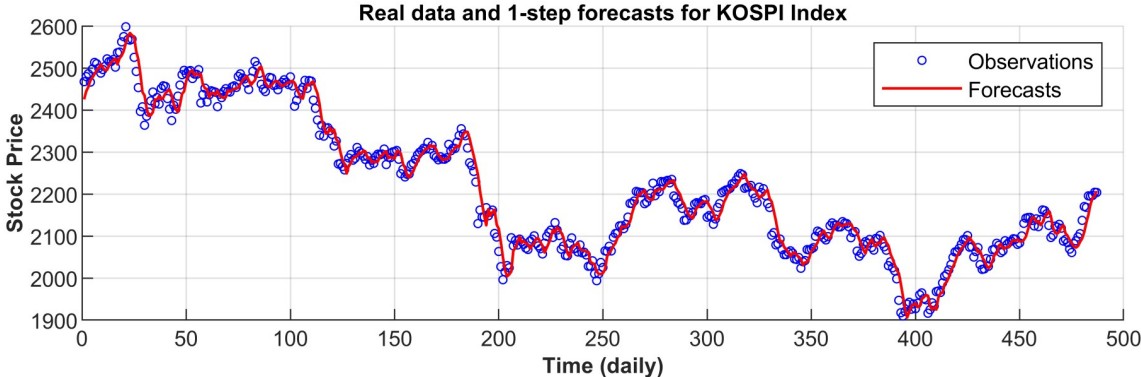

**Fig 5. 1-step forecasts of the nPRM and market values for Korea KOSPI.**

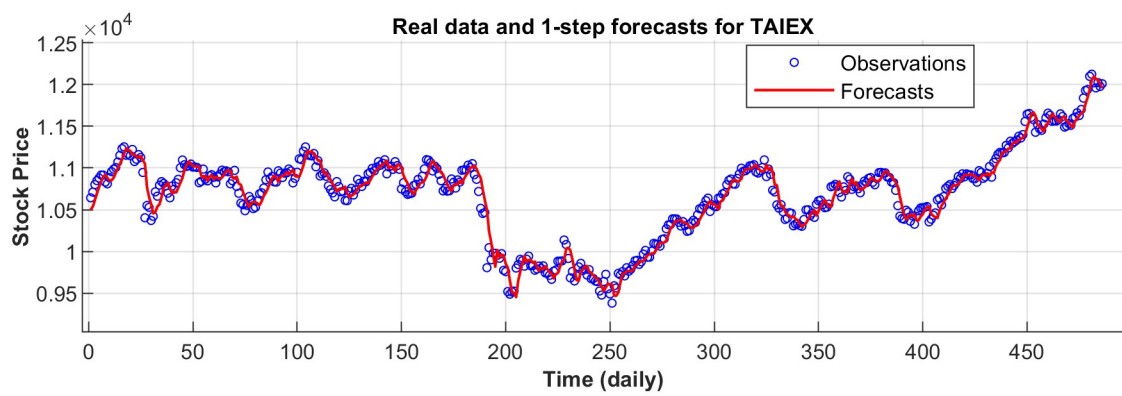

**Fig 6. 1-step forecasts of the nPRM and market values for Taiwan TAIEX.**

Apart from the martingale, we can investigate the level of market efficiency by sample entropy [38, 39]. Sample entropy measures complexity. When we have a time series of length $n$ as $\{S(1), S(2), \cdots, S(n)\}$, a template vector of length $m$ is defined as $S_m(i) = \{S(i), S(i+1), \cdots, S(i+m-1)\}$ and the distance function $d[S_m(i), S_m(j)], i \neq j$ is defined as the Chebyshev

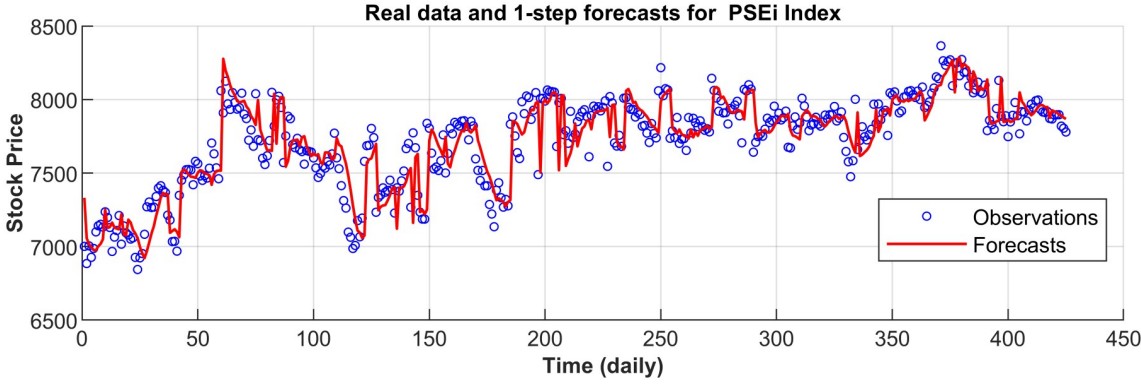

**Fig 7. 1-step forecasts of the nPRM and market values for the Philippines PSEi Index.**

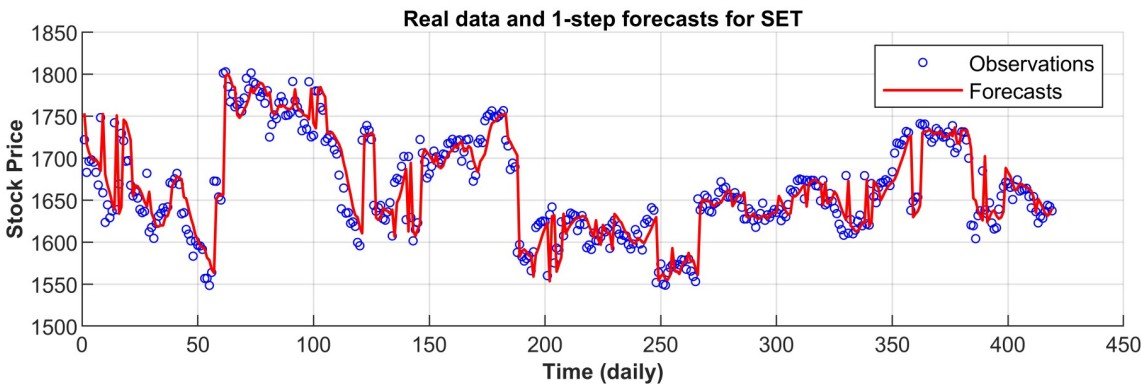

**Fig 8. 1-step forecasts of the nPRM and market values for Thailand SET Index.**

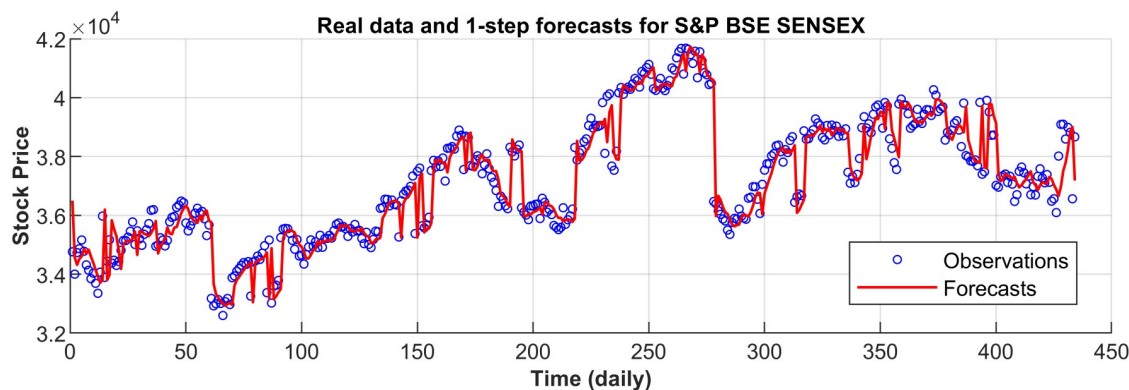

**Fig 9. 1-step forecasts of the nPRM and market values for India S&P BSE SENSEX.**

distance. Then, we define the sample entropy of the series, *SampeEn*, as

$$SampEn = -\log\frac{B^{m+1}}{B^m}, \tag{11}$$

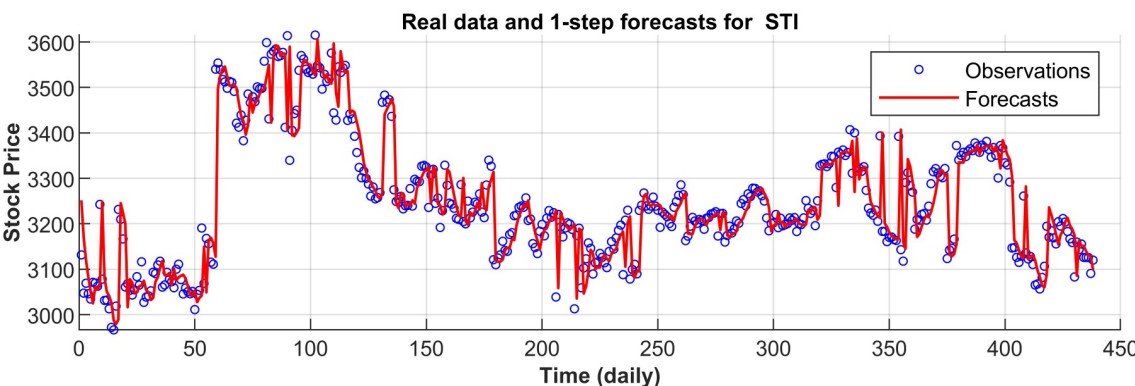

**Fig 10. 1-step forecasts of the nPRM and market values for Singapore Straits Times Index (STI).**

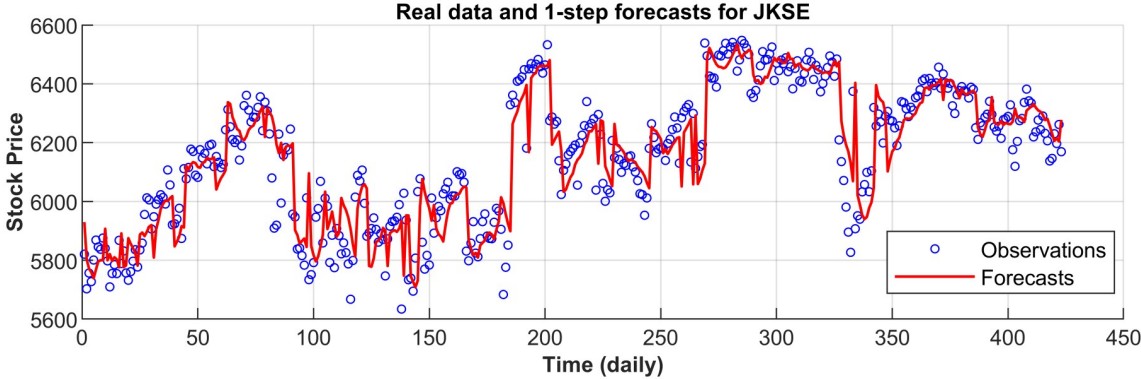

**Fig 11. 1-step forecasts of the nPRM and market values for Indonesia JKSE.**

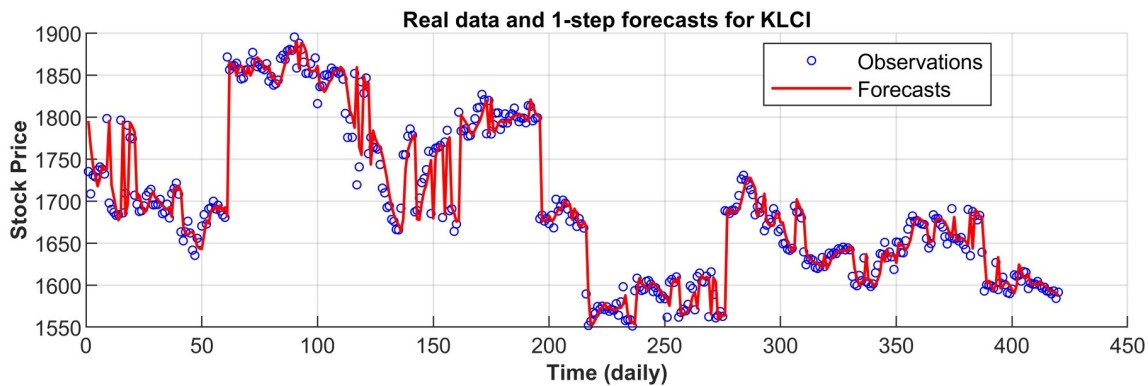

**Fig 12. 1-step forecasts of the nPRM and market values for Malaysia KLCI.**

where $B^m$ is number of template vector pairs having $d[S_m(i), S_m(j)] \leq r$, and $r$ is the tolerance. The value of sample entropy is nonnegative. The value 0 of sample entropy indicates that the series is perfectly regular without noise while a high value reflects randomness and unpredictability. If we examine the market efficiency from the perspective of sample entropy, a high value of sample entropy indicates a market with high level of efficiency. To show how large the possible value of sample entropy from a highly random series can be, we simulate a series of length 180 from a normal distribution with zero mean and unit variance for 1000 times. The average value of sample entropy is 2.21.

Table 5 shows the mean value of sample entropy calculated from all series used in estimation of $A$ and $\kappa$ to make forecasts dynamically in the testing set for ten indexes in our empirical study. The mean values range between 0.5952 (Korea KOSPI) and 0.7833 (the Philippines PSEi), which represent similar complexity among these markets. We suspect that the forecasting errors should be larger for markets with higher level of efficiency since it is harder to predict the series with random noise. By the order of the mean sample entropy values and the MAPEs for the ten stock indexes, the nPRM shows a tendency toward better forecasts for series with lower market efficiency. That is, the forecasting ability of the nPRM usually increases as the mean sample entropy decreases. However, there is an exception of Taiwan TAIEX. The mean sample entropy obtained from the TAIEX series is the second highest, so that we won't expect a smaller prediction error among the ten indexes. However, the MAPEs

**Table 3. MAPE (%) for the testing set.**

| | | $\tau = 1$ | $\tau = 2$ | $\tau = 3$ | $\tau = 4$ | $\tau = 5$ |
|---|---|---|---|---|---|---|
| **Nikkei 225** | nPRM | 1.04 | 1.27 | 1.53 | 1.78 | 2.05 |
| | Martingale | 0.72 | 1.05 | 1.28 | 1.52 | 1.74 |
| | Curve fitting | 3.05 | 3.32 | 3.60 | 3.87 | 4.15 |
| **Hang Seng** | nPRM | 1.12 | 1.44 | 1.77 | 2.11 | 2.43 |
| | Martingale | 0.81 | 1.21 | 1.53 | 1.79 | 2.00 |
| | Curve fitting | 2.33 | 2.52 | 2.71 | 2.89 | 3.06 |
| **TAIEX** | nPRM | 0.76 | 1.00 | 1.21 | 1.43 | 1.65 |
| | Martingale | 0.57 | 0.85 | 1.03 | 1.22 | 1.38 |
| | Curve fitting | 1.96 | 2.14 | 2.32 | 2.49 | 2.67 |
| **KOPSI** | nPRM | 0.86 | 1.10 | 1.38 | 1.62 | 1.84 |
| | Martingale | 0.61 | 0.92 | 1.17 | 1.39 | 1.56 |
| | Curve fitting | 2.54 | 2.76 | 2.98 | 3.20 | 3.42 |
| **PSEi** | nPRM | 1.46 | 1.71 | 1.93 | 2.16 | 2.35 |
| | Martingale | 0.75 | 1.02 | 1.22 | 1.42 | 1.58 |
| | Curve fitting | 3.09 | 3.36 | 3.65 | 3.93 | 4.22 |
| **SET** | nPRM | 1.18 | 1.47 | 1.70 | 1.98 | 2.20 |
| | Martingale | 0.46 | 0.69 | 0.86 | 1.01 | 1.13 |
| | Curve fitting | 2.08 | 2.25 | 2.43 | 2.60 | 2.78 |
| **BSE SENSEX** | nPRM | 1.31 | 1.62 | 1.88 | 2.20 | 2.42 |
| | Martingale | 0.53 | 0.78 | 0.99 | 1.17 | 1.33 |
| | Curve fitting | 2.29 | 2.51 | 2.72 | 2.94 | 3.17 |
| **STI** | nPRM | 1.19 | 1.52 | 1.77 | 2.09 | 2.32 |
| | Martingale | 0.54 | 0.76 | 0.91 | 1.06 | 1.19 |
| | Curve fitting | 1.76 | 1.91 | 2.05 | 2.18 | 2.31 |
| **JKSE** | nPRM | 1.27 | 1.42 | 1.57 | 1.73 | 1.84 |
| | Martingale | 0.60 | 0.85 | 1.02 | 1.14 | 1.27 |
| | Curve fitting | 2.39 | 2.62 | 2.85 | 3.08 | 3.31 |
| **KLCI** | nPRM | 0.98 | 1.30 | 1.58 | 1.89 | 2.07 |
| | Martingale | 0.49 | 0.72 | 0.90 | 1.05 | 1.18 |
| | Curve fitting | 1.76 | 1.92 | 2.07 | 2.22 | 2.38 |

of the nPRM for 1-step to 5-step forecasts are relatively low. The changes of TAIEX are similar to the movement which is characterized by the nPRM among the ten markets.

We summarize the performances of the methods as follows. Considering the deviation-type forecasting errors, the martingale seems to perform the best among the methods. According to the empirical evidence [2], the test based on the martingale can be used to examine the market efficiency of the stock market. Despite the absence of the test in this study, the low errors of the martingale for these ten Asian stock indexes imply that the markets tend to be efficient during the period of the data. The martingale suggests that no information we can obtain from the past to forecast the future stock prices. However, our proposed method nPRM still captures the trend of the stock movements quite well, and provides meaningful coefficients, $A$ and $\kappa$, as well under the circumstances. If applying the nPRM to the markets that are less efficient, the model coefficients may acquire more information with economic meanings and obtain more accurate forecasts than the martingale.

The average computational times of all the investigated methods are shown in Table 6. The martingale uses the least computational time while the curve fitting spends the most. The

**Table 4. Directional Symmetry (DS) (%) for the testing set.**

| | | $\tau = 1$ | $\tau = 2$ | $\tau = 3$ | $\tau = 4$ | $\tau = 5$ |
|---|---|---|---|---|---|---|
| **Nikkei 225** | nPRM | 48.97 | 50.00 | 50.17 | 51.03 | 50.51 |
| | Curve fitting | 47.11 | 46.60 | 45.92 | 46.77 | 47.45 |
| **Hang Seng** | nPRM | 49.06 | 47.19 | 48.38 | 46.85 | 46.17 |
| | Curve fitting | 48.81 | 48.47 | 48.81 | 49.66 | 51.69 |
| **TAIEX** | nPRM | 49.23 | 47.69 | 50.77 | 51.62 | 52.48 |
| | Curve fitting | 50.51 | 53.4 | 52.04 | 53.91 | 53.40 |
| **KOPSI** | nPRM | 47.10 | 48.98 | 49.49 | 49.15 | 52.90 |
| | Curve fitting | 49.92 | 48.05 | 49.92 | 47.71 | 46.35 |
| **PSEi** | nPRM | 50.19 | 51.15 | 48.47 | 49.24 | 48.09 |
| | Curve fitting | 47.04 | 47.58 | 45.16 | 45.43 | 45.97 |
| **SET** | nPRM | 48.84 | 49.03 | 48.65 | 50.97 | 49.03 |
| | Curve fitting | 49.57 | 47.84 | 49.86 | 50.72 | 51.01 |
| **BSE SENSEX** | nPRM | 44.84 | 45.03 | 46.90 | 51.41 | 51.41 |
| | Curve fitting | 48.28 | 48.01 | 49.07 | 51.19 | 54.11 |
| **STI** | nPRM | 46.93 | 48.98 | 49.53 | 51.77 | 52.33 |
| | Curve fitting | 50.92 | 51.18 | 49.34 | 51.44 | 52.76 |
| **JKSE** | nPRM | 49.43 | 51.72 | 51.72 | 50.19 | 50.77 |
| | Curve fitting | 47.03 | 46.76 | 44.86 | 44.86 | 42.97 |
| **KLCI** | nPRM | 48.94 | 46.82 | 46.24 | 47.78 | 50.67 |
| | Curve fitting | 49.72 | 47.84 | 44.84 | 46.53 | 43.71 |

**Table 5. Mean sample entropy for data series of length 180 used in calculation of $A$ and $\kappa$.**

| Nikkei 225 | Hang Seng | TAIEX | KOSPI | PSEi | SET | BSE SENSEX | STI | JKSE | KLCI |
|---|---|---|---|---|---|---|---|---|---|
| 0.6880 | 0.6094 | 0.7052 | 0.5952 | 0.7833 | 0.6130 | 0.6258 | 0.6593 | 0.6901 | 0.6088 |

nPRM is computationally efficient in view of the relationship between the forecasting accuracy and the computational time.

## Discussion

In this study, we propose a novel procedure adapted from the thermodynamic system to model the stock prices by a newly derived model, the new Price Reversion Model (nPRM), which is a modification of PRM. From the viewpoint of signal processing, the proposed procedure of nPRM models the stock prices with all available information. Therefore, the coefficients in the nPRM preserve and reveal more information about the stock movements economically. They also may be treated as important features for forecasting by advanced models such as deep learning models.

**Table 6. Average computational time (unit: Second).**

| | Nikkei 225 | Hang Seng | TAIEX | KOSPI | PSEi | SET | BSE SENSEX | STI | JKSE | KLCI |
|---|---|---|---|---|---|---|---|---|---|---|
| **nPRM** | 0.9715 | 0.8825 | 0.9584 | 0.9583 | 0.8191 | 0.8461 | 0.8695 | 1.0289 | 0.9276 | 0.9896 |
| **Martingale** | 0.0259 | 0.0230 | 0.0229 | 0.0269 | 0.0259 | 0.0229 | 0.0289 | 0.0239 | 0.0209 | 0.0189 |
| **Curve fitting** | 1.8553 | 1.7147 | 1.6480 | 1.7692 | 1.0994 | 1.0313 | 1.1555 | 1.0787 | 1.0677 | 1.5757 |

We evaluate the forecasting accuracy of the nPRM along with the martingale, and the cubic curve fitting by ten Asian stock index examples. By investigating the daily closing prices, the nPRM accurately characterizes and forecasts the large-scale motions in these ten stock markets. Although the forecasting accuracy declines over time, which is the same phenomenon for all the methods, the deviation-type errors of the nPRM are the smallest except for the martingale. However, the martingale can not be employed to forecast the trend of the stock even though it offers an eligible prediction about the prices if the market is efficient.

The trend-type errors of the nPRM and the curve fitting are close. Furthermore, the trend-type errors of the two methods do not significantly deviate from 0.5 from 1-step to 5-step forecasts. This indicates that the nPRM and the curve fitting does not possess good trend forecasting ability. In order to improve the trend forecasting in the future, we will develop a more sophisticated method to calculate the $\kappa$ values without the gradient control.

## Supporting information

**S1 Appendix. The martingale.**
(PDF)

**S2 Appendix. The curve fitting.**
(PDF)

**S1 Table. Correct up-trend (CP) (%) for the testing set.**
(PDF)

**S2 Table. Correct down-trend (CD) (%) for the testing set.**
(PDF)

## Acknowledgments

This research would not have been possible without the conceptual and theoretical development from Prof. Meng-Rong Li, the retired professor from Department of Mathematical Sciences, National Chengchi University. The authors are inspired by his works and have unreserved support from him.

## Author Contributions

**Data curation:** Tsung-Jui Chiang-Lin.

**Formal analysis:** Tsung-Jui Chiang-Lin, Yong-Shiuan Lee.

**Funding acquisition:** Shang-Yueh Tsai.

**Investigation:** Tsung-Jui Chiang-Lin, Yong-Shiuan Lee, Tzong-Hann Shieh.

**Methodology:** Tsung-Jui Chiang-Lin, Yong-Shiuan Lee, Tzong-Hann Shieh, Chien-Chang Yen, Shang-Yueh Tsai.

**Project administration:** Tzong-Hann Shieh.

**Resources:** Tsung-Jui Chiang-Lin, Yong-Shiuan Lee.

**Supervision:** Tzong-Hann Shieh, Shang-Yueh Tsai.

**Validation:** Tsung-Jui Chiang-Lin, Yong-Shiuan Lee, Tzong-Hann Shieh, Chien-Chang Yen.

**Visualization:** Tsung-Jui Chiang-Lin, Yong-Shiuan Lee.

**Writing – original draft:** Tsung-Jui Chiang-Lin, Yong-Shiuan Lee.

**Writing – review & editing:** Tzong-Hann Shieh, Chien-Chang Yen, Shang-Yueh Tsai.

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
