## [Decision Letter · Decision Letter 0]

10 May 2021

PONE-D-21-10180

Study of Asian Indexes by a Newly Derived Dynamic Model

PLOS ONE

Dear Dr. Shieh,

Thank you for submitting your manuscript to PLOS ONE. After careful consideration, we feel that it has merit but does not fully meet PLOS ONE’s publication criteria as it currently stands. Therefore, we invite you to submit a revised version of the manuscript that addresses the points raised during the review process.

We look forward to receiving your revised manuscript.

Kind regards,

Junhuan Zhang, PhD

Academic Editor

PLOS ONE

2. In your Methods section and Data availability statement, please clarify how the data used in this study were obtained, and how others can access these, such as by providing references, links or contact details to the original data source.

"No"

Reviewers' comments:

Reviewer's Responses to Questions

**Comments to the Author**

1. Is the manuscript technically sound, and do the data support the conclusions?

Reviewer #1: Yes

Reviewer #2: Yes

Reviewer #3: Yes

2. Has the statistical analysis been performed appropriately and rigorously? 

Reviewer #1: Yes

Reviewer #2: Yes

Reviewer #3: Yes

3. Have the authors made all data underlying the findings in their manuscript fully available?

Reviewer #1: No

Reviewer #2: Yes

Reviewer #3: Yes

4. Is the manuscript presented in an intelligible fashion and written in standard English?

Reviewer #1: Yes

Reviewer #2: Yes

Reviewer #3: Yes

5. Review Comments to the Author

Reviewer #1: Stockmarket index forecasting is a widely studied subject and will continue as long as the stock markets exist. This research has developed a new method combining different optimisation methods: Random Walk, a stochastic method with Dynamic programming to be applied on time series with uncertainties. It is an appreciated approach. But the article has to be improved by applying the following:

1. Format of the paper is weak, please correct.

2. Include the contribution of the paper in the abstract, Introduction and Conclusion

3. The is no background used at all. A method focused paper is not interesting for the investors. Please add a background on the stock market index forecasting and take them out of the Introduction

4. Please clearly indicate why you needed the PRN method instead of any other forecasting method like Gradient Boosting

5. Nunber of references have to be at least 30

Reviewer #2: The paper analyses the explanatory and predictive power of thermodynamic models in describing and forecasting the motion of broad-based stock market indexes. In particular, the authors suggest a new price reversion model (referred to as nPRM) aimed to overcome some limitations of classic price reversion models (PRM). In order to study the empirical performance of the nPRM, the authors use market data of four Asian stock indexes to calibrate the model and analyze forecasting errors as well as trend accuracy. The authors show that, while the PRM provides smaller errors for the 1-step forecast than the nPRM, the increase in the number of forecasting steps allows the nPRM to provide smaller errors than the PRM. On the other hand, in general, the PRM and nPRM have similar percentage of correct trend forecasts. Furthermore, the authors show that the nPRM exhibits a higher predictive power in forecasting the down-trends of market indexes.

In general, the paper is interesting and is well written. Moreover, the methodology is clearly described and the validation procedure is easily understandable. In any case, I believe that the authors should introduce some changes in the manuscript in order to allow the reader to better place their contributions in the current body of knowledge. Specifically:

1. In the recent decades, the literature on asset pricing has been prolific in providing us with new predictors that have increased substantially the predictive power of return forecasting models, especially in the case of stock market indexes. Thus, the classic view of financial market, where returns were almost unforecastable –consistent with the efficient-market hypothesis (Fama, 1970)–, has been clearly overcome by different models that exploit the strong predictive power of slow-moving predictors, such as the dividend yield, the price-to-earnings ratio, or the consumption-wealth ratio. Although this paper adopts a completely different perspective to analyze the stochastic behavior of stock indexes, I believe that the authors should better place their contribution in the current body of work, referring to important papers, such as Campbell and Shiller (1988ab, 2005), Lettau and Ludvigson (2001), Lewellen (2004) or Cochrane (2008). Cochrane (2011) provides a superb revision on the topic.

2. I believe that the main contribution of the paper and the usefulness of the proposed model should be better explained in the Introduction Section. Although the paper is clear and concise, in my opinion these aspects have not been sufficiently addressed.

3. In line with the previous comment, I believe that the explanations provided in the Results Section are excessively synthetical. The authors should make an effort to better describe the results and their implications, which is somewhat applicable to the Conclusions Section.

4. Although the manuscript is well-written, there are some typos that should be corrected (e.g., “the primary goals in this study is to identify” in line 36, etc.).

5. TAIEX is first mentioned in the paper in line 16, while the acronym is defined in 21. It is important that the authors correct these minor issues throughout the document.

REFERENCES:

Campbell, John Y., and Robert J. Shiller, 1988a, “The Dividend-Price Ratio and Expectations of Future Dividends and Discount Factors,” Review of Financial Studies 1, 195–228.

Campbell, John Y., and Robert J. Shiller, 1988b, “Stock Prices, Earnings, and Expected Dividends,” Journal of Finance 43, 661–676.

Cochrane, John H., 2008, “The Dog That Did Not Bark: A Defense of Return Predictability,” Review of Financial Studies 21, 1533–1575.

Cochrane, John H., 2011, “Presidential Address: Discount Rates,” Journal of Finance 66, 1047–1108.

Fama, Eugene F., 1970, “Efficient Capital Markets: A Review of Theory and EmpiricalWork,” Journal of Finance 25, 383–417.

Lettau, Martin, and Sydney C. Ludvigson, 2001a, “Consumption, Aggregate Wealth, and Expected Stock Returns,” Journal of Finance 56, 815–849.

Lewellen, Jonathan, 2004, “Predicting Returns with Financial Ratios,” Journal of Financial Economics 74, 209–235.

Reviewer #3: A new dynamic model for stock price index is proposed in this study and its validity is discussed in four Asian indexes. The manuscript is be accepted if it is revised well according to the comments.

1. Newly derived Price reversion model (nPRM) may be a revised version of original Price reversion model (PRM). As the background of this study, PRM should be introduced and nPRM is compared with PRM in order to reveal the their difference.

2. Martingale and Curve fitting are also described briefly because nPRM is compared with them.

3. Numerical examples reveal that PRM is better than nPRM at small value of ¥tau and that Martingale seems to be best among them. The validity of nPRM is not obvious. Authors should discuss these points.

4. Computational cost is also important point of view. The computational costs of four algorithms, nPRM, PRM, Martingale and Curve fitting, should be compared.

6. PLOS authors have the option to publish the peer review history of their article (what does this mean?). If published, this will include your full peer review and any attached files.

Reviewer #1: No

Reviewer #2: No

Reviewer #3: No

---

## [Author Response · Author response to Decision Letter 0]

15 Sep 2021

Submission No: PONE-D-21-10180

Dear Editor,

 We appreciate the reviewers’ valuable comments and suggestions. Meanwhile, we have addressed those comments one by one in the following responses. In the following, the original comments from the reviewer are in black and our responses are in blue.

With Regards,

Yong Shiuan, Lee

Response to Comments from Reviewer #1

1. Format of the paper is weak, please correct.

We rearrange the outline of this article. We follow the submission guidelines of the “Manuscript Organization” of PLOS ONE. We present the sections/subsections as follows. 

Abstract

Introduction

Methods

 � The Price Reversion Model (PRM)

 � Derivation of solutions to the new Price Reversion Model (nPRM)

 � Implementation details of the nPRM

 � Model evaluation

Results

Simulation design

Applied data

Simulation results

Discussion

Acknowledgments

References

Supporting information captions

 � S1 Appendix - The martingale approach

 � S2 Appendix - The curve fitting approach

 � S3 Fig.

 � S4 Fig.

 � S5 Fig.

 � S6 Fig.

2. Include the contribution of the paper in the abstract, Introduction and Conclusion.

We rewrite the Abstract, the Introduction Section, and the Conclusion Section. The distinguishing features and the contribution of the proposed model are presented in the end of the Introduction Section. The contribution of our work is included in the Abstract, the Introduction Section, and the Conclusion Section.

In details:

Abstract (the last two sentences): 

“This proposed procedure brought a different perspective to the study of stock prices. The time varying coefficients in the nPRM offer economic meanings of the stock movements.”

Introduction Section (line 90-97):

“The contributions of this paper are in the following.

The proposed procedure of nPRM directly analyzes raw data of stock prices, which retains all available information up to date to avoid reducing low-frequency information by taking returns or the logarithm of returns as well as making assumptions about the model. Moreover, the coefficients in the nPRM, namely the equilibrium price and the gradient, have their very own economic meanings. The gradient value reflects the convergence speed to the equilibrium price. We may further utilize the series of the gradient values as a feature for other models.”

Discussion Section (line 382-386):

“From the viewpoint of signal processing, the proposed procedure of nPRM models the stock prices with all available information. Therefore, the coefficients in the nPRM preserve and reveal more information about the stock movements economically. They also may be treated as important features for forecasting by advanced models such as deep learning models.”

3. The is no background used at all. A method focused paper is not interesting for the investors. Please add a background on the stock market index forecasting and take them out of the Introduction.

 We add some background of stock market. We also rewrite the Introduction Section to review approaches of stock forecasting from different perspectives, such as financial accounting, econometrics, signal processing plus artificial intelligence models, differential equations and dynamic systems.

In details:

Introduction Section (line 3-68): 

“The market prices of an individual stock reflect the company's existing value or future profit growth. A stock index, measured by the weighted mean prices of selected stocks in a stock market, represents the stock market's performance and potential competitive ability. Investors who have different motives for participating in the market concern about the changes of a stock or a stock index's market prices of different scales, intraday high-frequency data, daily closing prices, weekly or monthly summary, and etc. Accurately forecasting the stock movements could help the investors with decisions about investment strategy.

Whether the stock prices can be forecasted has been repeatedly discussed in finance. The efficient market hypothesis forms a foundation of the theory that under what circumstances the stock prices is predictable. The efficient market hypothesis states that three types of markets can be distinguished as the weak form, the semi-strong form, and the strong form depending on the level of market efficiency. The stock prices fully reflect all available market information in a strong efficient market, thus they are not predictable. The martingale depicts the stock movements in the strong efficient markets by the theory of random walks. In contrast, many real markets are considered as the weak or the semi-strong efficient markets. Consequently, to forecast the prices in these markets is possible.

The analysis of the stock movements is based on either the stock prices or the stock returns. The choice between the stock prices and the returns mainly depends on the approach adopted. …”

4. Please clearly indicate why you needed the PRM method instead of any other forecasting method like Gradient Boosting.

 The gradient boosting is an ensemble method which combines weak prediction models (classification, regression, etc.) to form a predictive model. This is a well known and commonly used technique in machine learning. We are really grateful for this valuable comment although we did not involve this in the previous manuscript. On the contrary, our proposed model takes a different perspective by using the differential equations and the dynamic systems. We choose the martingale as a baseline/benchmark to compare with the nPRM if the market is efficient. Additionally, we choose the curve fitting to compare with the nPRM since in our proposed procedure the coefficients are determined piecewisely (constant in each window). The coefficients change with time (window by window). Therefore, the dynamics of the coefficients characterizes the stock movements. This procedure is similar to the approach of curve fitting, which solves the parameters by OLS piewisely. Those parameters also change in different intervals. Since the PRM and the nPRM are deterministic models, which solve the differential equations in an interval [t0, t1], it is hard for us to apply the gradient boosting during our proposed procedure or compare the nPRM with it. However, we think that the gradient boosting may be implemented to find the best values of the coefficients (κ and A) in the nPRM in the future. Comparing the results of the nPRM and a boosted predictive model such as combining SVRs (support vector regression) may also be the future work.

5. Number of references has to be at least 30.

 We add some references as follows. Cowles (1933), Abarbanell and Bushee (1997, 1998), Park and Irwin (2007), Malkiel and Fama (1970), Campbell and Shiller (1988 [9]; 1988 [12]; 2005), Lettau and Ludvigson (2001), Lewellen (2004), Cochrane (2008; 2011), Welch and Goyal (2008), Rapach et al. (2010), Cheng and Wei (2014), Lim and Luo (2012), Wang and Wang (2015), Zhang and Pan (2015), Yan et al. (2020), Basak et al. (2019), Xu et al. (2016), Mallikarjuna and Rao (2019). We also brief important references in the Related Work Section.

In details:

The number of references is 38 now. 

Response to Comments from Reviewer #2

1. In the recent decades, the literature on asset pricing has been prolific in providing us with new predictors that have increased substantially the predictive power of return forecasting models, especially in the case of stock market indexes. Thus, the classic view of financial market, where returns were almost unforecastable –consistent with the efficient-market hypothesis (Fama, 1970)–, has been clearly overcome by different models that exploit the strong predictive power of slow-moving predictors, such as the dividend yield, the price-to-earnings ratio, or the consumption-wealth ratio. Although this paper adopts a completely different perspective to analyze the stochastic behavior of stock indexes, I believe that the authors should better place their contribution in the current body of work, referring to important papers, such as Campbell and Shiller (1988ab, 2005), Lettau and Ludvigson (2001), Lewellen (2004) or Cochrane (2008). Cochrane (2011) provides a superb revision on the topic.

 We add some references as follows. Cowles (1933), Abarbanell and Bushee (1997, 1998), Park and Irwin (2007), Malkiel and Fama (1970), Campbell and Shiller (1988 [9]; 1988 [12]; 2005), Lettau and Ludvigson (2001), Lewellen (2004), Cochrane (2008; 2011), Welch and Goyal (2008), Rapach et al. (2010), Cheng and Wei (2014), Lim and Luo (2012), Wang and Wang (2015), Zhang and Pan (2015), Yan et al. (2020), Basak et al. (2019), Xu et al. (2016), Mallikarjuna and Rao (2019). We also brief important references in the Introduction Section.

In details:

Introduction Section (line 11-80) (partly): 

“Whether the stock prices can be forecasted has been repeatedly discussed in finance (Cowles1933, Lim2012). The efficient market hypothesis (Malkiel1970) forms a foundation of the theory that under what circumstances the stock prices is predictable.”,

“In contrast, many real markets are considered as the weak or the semi-strong efficient markets (Cochrane2011, Lim2012).”,

“The analysis of the stock movements is based on either the stock prices or the stock returns. The choice between the stock prices and the returns mainly depends on the approach adopted. From the financial accounting perspective, fundamental analysis (Abarbanell1997, Abarbanell1998) and technical analysis (Park2007) are the two main intellectual traditions.”,

“When using ARIMA and VAR models, the stationarity of the time series is required. Therefore, the stock returns instead of the stock prices are the primary research subject (Campbell1988b, Lewellen2004, Welch2008). Financial ratios like the log dividend-price ratio (Campbell1988[9], Campbell2005), the dividends (Campbell1988[12]), the price–earnings ratio (Campbell2005, Lewellen2004), the dividend yield (Cochrane2008, Lewellen2004), the consumption-wealth ratio (Lettau2001), the book-to-market ratio (Lewellen2004) serve as important predictors for asset returns in regression models, particularly VAR models. Although some good predictors in financial literature have restricted predictive ability for the equity premium due to model uncertainty and instability (Welch2008), combining individual forecasts produce out-of-sample gains (Rapach2010). Modeling stock returns reduces low-frequency information, the trend for example, from the perspective of signal processing. Therefore, forecasting the stock prices based on models constructed from detrended data may be less accurate in this sense.”,

“The artificial intelligence models applied to the stock prices in existing studies include support vector regression (SVR) (Cheng2014), neural networks (Atsalakis2009, Guresen2011, Wang2015, Zhang2015, Yan2020), tree-based models (Basak2019), etc. To applying the artificial intelligence models, acquiring features is an important step. The empirical mode decomposition (EMD) (Cheng2014, Zhang2015), the ensemble empirical mode decomposition (EEMD) (Xu2016), and the complementary ensemble empirical mode decomposition (CEEMD) (Yan2020) are widely employed methods to decompose the stock prices, which are regarded as signals. The empirical results of US and UK stock indexes show that the artificial intelligence models are better than the ARIMA and GARCH models in terms of forecasting directional symmetry (Zhang2015). Furthermore, the CEEMD-PCA-LSTM model (Yan2020), which integrate the CEEMD, principal component analysis (PCA), and long short-term memory (LSTM) networks, outperforms naive recurrent neural networks (RNN) or LSTM networks in terms of forecasting accuracy for six selected stock indexes. Still, in a review study of the returns from financial perspective (Mallikarjuna2019), the empirical evidences show that no single model out of statistical, artificial intelligence, frequency domain, and hybrid models under investigation could be applied uniformly to all markets. On the other hand, the direction prediction problem is another task, which can be tackled by the tree-based models (Basak2019) to predict the stock price direction.”,

 “Another study (Xu2016) also considers the stock markets as complex dynamical systems, but their interactions are analyzed by methods of signal processing, the EMD and the EEMD along with detrended cross-correlation analysis (DCCA).”,

2. I believe that the main contribution of the paper and the usefulness of the proposed model should be better explained in the Introduction Section. Although the paper is clear and concise, in my opinion these aspects have not been sufficiently addressed.

We rewrite the Introduction Section. The distinguishing features and the contribution of the proposed model are presented in the end of the Introduction Section.

In details:

Introduction Section (line 90-97):

“The contributions of this paper are in the following.

The proposed procedure of nPRM directly analyzes raw data of stock prices, which retains all available information up to date to avoid reducing low-frequency information by taking returns or the logarithm of returns as well as making assumptions about the model. Moreover, the coefficients in the nPRM, namely the equilibrium price and the gradient, have their very own economic meanings. The gradient value reflects the convergence speed to the equilibrium price. We may further utilize the series of the gradient values as a feature for other models.”

3. In line with the previous comment, I believe that the explanations provided in the Results Section are excessively synthetical. The authors should make an effort to better describe the results and their implications, which is somewhat applicable to the Conclusions Section.

We amend the Results Section with some more discussions. We also rewrite the Discussion Section (Conclusion Section in previous version of manuscript) to highlight the contribution of this article and point out possible future work.

In details:

Result Section (line 308-372) (important part):

“From the deviation-type errors, the martingale (see details in S1 Appendix) has the smallest errors followed by those from the nPRM, the PRM and the curve fitting (see details in S2 Appendix) in the ascending order. Comparing the nPRM with the PRM, although the PRM obtains smaller 1-step forecast errors, the PRM obtains larger errors than the nPRM with the increase of the number of forecasting steps. From the overall performance, the nPRM is more accurate than the PRM. The PRM seems to perform better at the 1-step forecast since the miscalculations for cases 3 and 4 of the nPRM in Table 1 do not lead to large errors. However, with the increases of the number of forecasting steps, those miscalculations of theoretical values accumulate errors.”

“The trend-type errors cannot be applied to the forecasts of the martingale since the martingale forecasts the price of the next day by the present price. Therefore, we compare the trend-type errors between the other three methods. The PRM, nPRM, and the curve fitting have similar percentage of correct trend forecasts. Although the differences between the three methods are not significant, the PRM produces slightly higher trend accuracy among these three methods.”

“We summarize the performances of the methods as follows. Considering the deviation-type forecasting errors, the martingale seems to perform the best among the methods. According to the empirical evidence (Lim2012), the test based on the martingale can be used to examine the market efficiency of the stock market. Despite the absence of the test in this study, the low errors of the martingale for the four Asian stock indexes imply that the markets tend to be efficient during the period of the data. The martingale suggests that no information we can obtain from the past to forecast the future stock prices. Under the circumstances, our proposed method nPRM captures the trend of the stock movements quite well, and provides meaningful coefficients, A and κ, as well. If applying the nPRM to the markets that are less efficient, the model coefficients may acquire more information with economic meanings and obtain more accurate forecasts than the martingale.

The PRM has some flaw in the integration process and does not require smoothing for the stock prices. These disadvantages cause errors in forecasts. The errors are small when the time interval of forecast is short, but they grow with the increase of the time length. The newly proposed method nPRM improves these weaknesses. Therefore, the forecasting errors are notably smaller than the PRM for multiple step forecasts.

By the deviation-type measurements, the PRM or nPRM outperforms the curve fitting while the martingale cannot deliver a forecast of trend.

In spite of small differences, the PRM has the highest forecasting accuracy in all of the four Asian markets, and produces better directional forecasts for the up trend with higher CP values when the nPRM has higher CD values. In brief, the trend forecasting ability of the three methods are close.”

Discussion Section (line 387-399):

“We evaluate the forecasting accuracy of the nPRM along with the PRM, the martingale, and the cubic curve fitting by four Asian stock index examples. By investigating the daily closing prices, the nPRM accurately characterizes and forecasts the large-scale motions in these four stock markets. Although the forecasting accuracy declines over time, which is the same phenomenon for all the methods, the deviation-type errors of the nPRM are the smallest except for the martingale. However, the martingale can not be employed to forecast the trend of the stock even though it offers an eligible prediction about the prices if the market is efficient.

The trend-type errors of the nPRM, the PRM, and the curve fitting are close. This indicates that the trend forecasting ability of the nPRM does not outperform the other methods as its overall forecasting ability. In order to improve the trend forecasting in the future, we will develop a more sophisticated method to calculate the κ values without the gradient control.”

4. Although the manuscript is well-written, there are some typos that should be corrected (e.g., “the primary goals in this study is to identify” in line 36, etc.).

 We corrected the typo in line 86 (line 36 in the previous version).

We changed the “is” to “are”.

All the expressions, “stock movement”, are changed as “stock movements” for consistency throughout the paper (three times in the Abstract and in line 9, 17, 21, 31, 80, 205, 248, 358, 385).

All the expressions, “stock price”, are changed as “stock prices” for consistency throughout the paper.

 We corrected the typos for the solutions of the PRM in line 126-133.

Cases A, B, C, D and equations (2), (3), (4), (5).

 We corrected the typos for the solutions of the nPRM.

S0 < A (line 162), S0 > A (line 166), S0 < A (line 170), increasing (line 171), S(t) < A (line 171), S0 > A (line 174), decreasing (line 175), S(t) > A (line 176).

5. TAIEX is first mentioned in the paper in line 16, while the acronym is defined in 21. It is important that the authors correct these minor issues throughout the document.

The acronym is defined when first mentioning TAIEX in line 74 (in the revised version).

Response to Comments from Reviewer #3

1. Newly derived Price reversion model (nPRM) may be a revised version of original Price reversion model (PRM). As the background of this study, PRM should be introduced and nPRM is compared with PRM in order to reveal their difference.

 We add a subsection to describe the original Price Reversion Model (PRM) in the Methods Section. We also add Table 1 to compare the solutions of the PRM with the solutions of the nPRM.

In details: 

Methods Section – The Price Reversion Model (PRM) subsection (line 122-136)

Methods Section - Derivation of solutions to the new Price Reversion Model (nPRM) (line 177-184)

“A comparison of the cases on different conditions between PRM and nPRM is given by Table 1. As mentioned before, cases B and D in the PRM do not exist due to zero external force at A. By examining the phase plots of Figure 1 and Figure 2, the case A of the PRM can be further divided into cases 2 and 4 of the nPRM. The case C of the PRM can be further divided into cases 1 and 3 of the nPRM. We derive solutions for the cases 3 and 4 of the nPRM, which can not be obtained identically from the solutions of the PRM. As a result, the theoretical values obtained via the PRM are not so accurate as those from the nPRM.”

2. Martingale and Curve fitting are also described briefly because nPRM is compared with them.

 We add Appendix S1 and S2 to describe the Martingale and the Curve fitting.

In details: 

We mention S1 Appendix and S2 Appendix in line 316 and 317.

3. Numerical examples reveal that PRM is better than nPRM at small value of tau and that Martingale seems to be best among them. The validity of nPRM is not obvious. Authors should discuss these points.

We compare the solutions of the PRM with the solutions of the nPRM by Table 1. We also add some paragraphs of discussion in the last part of the Results Section.

In details: 

Derivation of solutions to the new Price Reversion Model (nPRM) Section (line 177-184):

“A comparison of the cases on different conditions between PRM and nPRM is given by Table 1. As mentioned before, cases B and D in the PRM do not exist due to zero external force at $A$. By examining the phase plots of Figure 1 and Figure 2, the case A of the PRM can be further divided into cases 2 and 4 of the nPRM. The case C of the PRM can be further divided into cases 1 and 3 of the nPRM. We derive solutions for the cases 3 and 4 of the nPRM, which can not be obtained identically from the solutions of the PRM. As a result, the theoretical values obtained via the PRM are not so accurate as those from the nPRM.”

Results Section (line 317-324):

“Comparing the nPRM with the PRM, although the PRM obtains smaller 1-step forecast errors, the PRM obtains larger errors than the nPRM with the increase of the number of forecasting steps. From the overall performance, the nPRM is more accurate than the PRM. The PRM seems to perform better at the 1-step forecast since the miscalculations for cases 3 and 4 of the nPRM in Table 1 do not lead to large errors. However, with the increases of the number of forecasting steps, those miscalculations of theoretical values accumulate errors.”

4. Computational cost is also important point of view. The computational costs of four algorithms, nPRM, PRM, Martingale and Curve fitting, should be compared.

 We add Table 7 to present average computational time of the four methods in the empirical study. We discuss the comparison in the last paragraph in the Results Section.

In details: 

Results Section (line 373-378):

“The average computational times of all the four methods are investigated through 100 simulations. The average CPU times are shown in Table 7. The martingale uses the least computational time while the curve fitting spends the most. The nPRM and the PRM take nearly the same time, which is approximately half of the time used by the curve fitting. Even so, the nPRM is computationally efficient in view of the relationship between the forecasting accuracy and the computational time.”

---

## [Decision Letter · Decision Letter 1]

9 Nov 2021

PONE-D-21-10180R1Study of Asian indexes by a newly derived dynamic modelPLOS ONE

Dear Dr. Lee,

Thank you for submitting your manuscript to PLOS ONE. After careful consideration, we feel that it has merit but does not fully meet PLOS ONE’s publication criteria as it currently stands. Therefore, we invite you to submit a revised version of the manuscript that addresses the points raised during the review process.

We look forward to receiving your revised manuscript.

Kind regards,

Junhuan Zhang, PhD

Academic Editor

PLOS ONE

Reviewers' comments:

Reviewer's Responses to Questions

**Comments to the Author**

1. If the authors have adequately addressed your comments raised in a previous round of review and you feel that this manuscript is now acceptable for publication, you may indicate that here to bypass the “Comments to the Author” section, enter your conflict of interest statement in the “Confidential to Editor” section, and submit your "Accept" recommendation.

Reviewer #3: All comments have been addressed

Reviewer #4: (No Response)

2. Is the manuscript technically sound, and do the data support the conclusions?

Reviewer #3: Partly

Reviewer #4: Partly

3. Has the statistical analysis been performed appropriately and rigorously? 

Reviewer #3: I Don't Know

Reviewer #4: Yes

4. Have the authors made all data underlying the findings in their manuscript fully available?

Reviewer #3: Yes

Reviewer #4: Yes

5. Is the manuscript presented in an intelligible fashion and written in standard English?

Reviewer #3: Yes

Reviewer #4: Yes

6. Review Comments to the Author

Reviewer #3: The manuscript is revised according to the reviewer's comment.

However, the section titles should be numbered adequately for readers' convenience.

Reviewer #4: The authors propose a new Price Reversion Model (nPRM), which is adapted from the Newton’s law of cooling. The solution of the model is derived and is discussed under different circumstances. Then nPRM is applied to forecast market values of four Asian stock indexes. The results do not seem so convincing. The work is meaningful to the research of stock market, but there are some modifications.

Major:

(1) The abstract is too cumbersome and a bit long. Especially the first four sentences are somewhat redundant.

(2) In the introduction, the authors introduced a variety of methods for predicting stock prices, and commented that there is no single method that can be used in all markets. I think this evaluation is not objective. No method is universally applicable to all markets, including the method proposed in this article, which has poor predictive effect in efficient markets. Therefore, what are the differences between the nPRM and the existing methods, or what are the advantages of the nPRM in predicting prices? Please elaborate on the importance of proposing this method.

(3) The authors put forward two primary goals for PRM in the introduction. However, the contribution of PRM to price prediction does not seem to be expressed. Does PRM have a large impact on the academic field or is it worthy of being an improvement target? The second goal does not seem to have been achieved.

(4) The first contribution cannot be considered a contribution. If you don't care about the prediction accuracy, all models can input raw data. But whether this is right is worth discussing.

(5) In the model part, what is the economic implications or intuition of nPRM? Why is the change in price assumed to be such a differentiable equation?

(6) The authors explain that nPRM is better than PRM because the former has two more cases than the latter. How do the authors identity it? Why is it not because smooth historical data is used when calculating A? Why is it not caused by controlling K?

(7) In nPRM, A and K are the most critical parameters, and their estimation method will directly determine the predicted performance. Authors should focus on how to estimate A and K. How to determine the optimal A and K is the key. From the prediction results, it is unreasonable to assume that the index obeys geometric Brownian motion.

(8) In the results discussion, authors say that if applying the nPRM to the markets that are less efficient, the model coefficients may acquire more information with economic meanings and obtain more accurate forecasts than the martingale. What is the reason for this explanation? What is the definition of market efficiency? How is the author identified? If nPRM performs better in an inefficient market, why don't the authors consider using other indexes? If nPRM can perform better in other samples, I will think that this method is meaningful and contributing.

7. PLOS authors have the option to publish the peer review history of their article (what does this mean?). If published, this will include your full peer review and any attached files.

Reviewer #3: No

Reviewer #4: No

---

## [Author Response · Author response to Decision Letter 1]

20 Dec 2021

Submission No: PONE-D-21-10180

Dear Editor,

 We appreciate the reviewers’ valuable comments and suggestions. Meanwhile, we have addressed those comments one by one in the following responses. In the following, the original comments from the reviewer are in black and our responses are in blue.

With Regards,

Yong Shiuan, Lee

Response to Comments from Reviewer #3

1. The manuscript is revised according to the reviewer's comment.

However, the section titles should be numbered adequately for readers' convenience.

 We agree with this comment that numbered sections/subsections are more readable. As a matter of fact, we follow the submission guidelines of the “Manuscript Organization” of PLOS ONE, and organize the sections by using the downloaded sample file. The format of the template for LaTeX submission does not number the sections and subsections. Therefore, we cannot present the manuscript with numbered section titles.

We present the revised manuscript as follows. 

Abstract

Introduction

Methods

 � The Price Reversion Model (PRM)

 � Derivation of solutions to the new Price Reversion Model (nPRM)

 � Implementation details of the nPRM

 � Model evaluation

Results

Applied data

Empirical results

Discussion

Acknowledgments

References

Supporting information captions

 � S1 Appendix - The martingale approach

 � S2 Appendix - The curve fitting approach

 � S3 Table Correct up-trend (CP) (%) for the testing set.

 � S4 Table Correct down-trend (CD) (%) for the testing set.

Response to Comments from Reviewer #4

1. The abstract is too cumbersome and a bit long. Especially the first four sentences are somewhat redundant.

 We exclude the first four sentences in the abstract. We also amend and rearrange the sentences in the abstract since we add more empirical examples.

In details:

Abstract (from the 5th line in the abstract):

“…This proposed procedure brought a different perspective to the study of stock prices based on thermodynamics, and the time varying coefficients in the nPRM offer economic meanings of the stock movements. More specifically, the average of smoothed historical data A in the nPRM, analogous to the environment temperature in the Newton's law of cooling, represent an implied equilibrium price. The heat transfer coefficient k is adapted to be either negative or positive, which illustrates the speed of convergence or divergence of stock prices, respectively. The empirical study of ten Asian stock indexes shows that the nPRM accurately characterizes and forecasts the market values.”

2. In the introduction, the authors introduced a variety of methods for predicting stock prices, and commented that there is no single method that can be used in all markets. I think this evaluation is not objective. No method is universally applicable to all markets, including the method proposed in this article, which has poor predictive effect in efficient markets. Therefore, what are the differences between the nPRM and the existing methods, or what are the advantages of the nPRM in predicting prices? Please elaborate on the importance of proposing this method.

In the previous edition of our manuscript (line 63-66), we summarize research results from reference [25] and state that “Still, in a review study of the returns from financial perspective [25], the empirical evidences show that no single model out of statistical, artificial intelligence, frequency domain, and hybrid models under investigation could be applied uniformly to all markets.” We would like to point out that disadvantage of modeling returns instead of stock prices is a loss of low-frequency information by the signal processing perspective. That’s why we choose to model daily closing prices in this work.

The proposed method nPRM produces predictions with relatively low Mean Absolute Percentage Errors (MAPEs) in our empirical study compared to other methods regardless of the market efficiency. More importantly, the difference in MAPE between nPRM and the martingale is little. 

Indeed, “all models are wrong, but some are useful”, said by George Box. This quotation refers to not only statistical models but also other scientific models. However, we are dedicated to construct models that 1. have logical representation and economically reasonable explanations, and 2. produce predictions as accurately as possible.

The main contributions of this proposed method include describing the stock price movement accurately and provide economically meaningful coefficients at the mean time.

3. The authors put forward two primary goals for PRM in the introduction. However, the contribution of PRM to price prediction does not seem to be expressed. Does PRM have a large impact on the academic field or is it worthy of being an improvement target? The second goal does not seem to have been achieved.

The PRM is developed upon the Newton’s law of cooling, which is a well known thermodynamic model and introduced into finance in some researches, for example, 

“Gkranas, A., Rendoumis, V.L. and Polatoglou, H.M. (2004). Athens and Lisbon stock markets – a thermodynamic approach, WSEAS Transactions on Business and Economics, Vol. 1, No. 1, pp.95–100”,

“Zarikas, V., Christopoulos, A.G. and Rendoumis, V.L. (2009). A thermodynamic description of the time evolution of a stock market index, European Journal of Economics, Finance and Administrative Sciences, Vol. 16, pp.73–83”,

“Todorović, J. Đ., Tomić, Z., Denić, N., Petković, D., Kojić, N., Petrović, J., & Petković, B. (2018). Applicability of Newton’s law of cooling in monetary economics. Physica A - Statistical Mechanics and its Applications, 494, 209-217”,

and our reference [27] among others.

 Although PRM might not have brought great impact to the academic field of finance, the concept of introducing the thermodynamics into stock index dynamic modeling is new and draws some attention. This approach models the stock prices from a different perspective without loss of information due to transformation or model assumptions.

 We slightly amend the introduction, especially the paragraph detailing the primary goals and contributions of this work (line 85-90).

In details:

Abstract (line 5-12):

“This proposed procedure brought a different perspective to the study of stock prices based on thermodynamics, and the time varying coefficients in the nPRM offer economic meanings of the stock movements. More specifically, the average of smoothed historical data A in the nPRM, analogous to the environment temperature in the Newton's law of cooling, represent an implied equilibrium price. The heat transfer coefficient k is adapted to be either negative or positive, which illustrates the speed of convergence or divergence of stock prices, respectively. The empirical study of ten examples shows that the nPRM accurately characterizes and forecasts the market values for Asian stock indexes.”

Introduction (line 85-94):

“Therefore, we identify the reasons of inaccurate forecasts by the PRM and propose to model the stock prices by a newly derived model, i.e. the new Price Reversion Model (nPRM). The contributions of this paper are in the following. The proposed procedure of nPRM directly analyzes raw data of stock prices, which retains all available information up to date to avoid reducing low-frequency information by taking returns or the logarithm of returns as well as making assumptions about the model. Moreover, the coefficients in the nPRM, namely the equilibrium price and the gradient, have their very own economic meanings. The gradient value reflects the convergence speed to the equilibrium price. We may further utilize the series of the gradient values as a feature for other models.”

4. The first contribution cannot be considered a contribution. If you don't care about the prediction accuracy, all models can input raw data. But whether this is right is worth discussing.

 The first contribution is “directly analyzes raw data of stock prices, which retains all available information up to date to avoid reducing low-frequency information …” (line 87-90). Since a great part of recent researches of stock in academic field of finance is based on stock returns, we try to describe the stock price movement directly without loss of information from the signal processing perspective. We try to construct a reasonable model, which characterizes our subject properly and creates precise predictions.

 We do care about the prediction accuracy, which is mainly measured by the Mean

Absolute Percentage Error (MAPE) in this article. Empirical study shows that the prediction accuracy of nPRM is really close to the martingale.

In the previous edition of our manuscript, the measurement trend accuracy (Acc) is used to determine whether the direction of the prediction is the same as real market value. This measurement along with CD and CP are useful in practice if we focus on the direction which stock prices go. For a martingale, either direction happens with a 50/50 chance. We replace Acc by another terminology, Directional Symmetry (DS), in our revised manuscript to avoid misunderstanding. Larger values of DS, CD, and CP represent more price predictions going in the same direction with real market.

In details:

Methods-Model evaluation (line 236-237, 244-249):

“directional Symmetry (DS)”,

“A smaller value of the MAPE indicates that the model better describes the movements of the stock prices. The MAE and the RMSE show similar tendency with the MAPE. DS measures the frequency of correct directional forecasts. CP and CD measure the frequency of correct directional forecasts when real market values rise and fall, respectively. The larger values of DS, CP, and CD suggest that the model obtains more correct directional forecasts in general, for uptrend, and for downtrend, respectively.”

5. In the model part, what is the economic implications or intuition of nPRM? Why is the change in price assumed to be such a differentiable equation?

The economic implication of nPRM is briefly described in the Section Derivation of solutions to the new Price Reversion Model (nPRM) (line 136-150). We refer A to the market equilibrium price. We refer k to the speed of convergence or divergence of stock prices, which is determined by the force on stock. 

We assume that the dynamics of the price movement is differentiable (at least twice differentiable) to time t. However, in reality we observe the price at discrete timepoints. In this study, the observations are made once a day (daily closing prices). Since we cannot observe the price at an infinity of possible argument time t, we need smoothing technique to obtain differentiable curves that capture important trend and features of our data (step 1 in the implementation of nPRM). This smoothing step is neglected in PRM so that this modification in nPRM makes it more theoretically reasonable.

6. The authors explain that nPRM is better than PRM because the former has two more cases than the latter. How do the authors identity it? Why is it not because smooth historical data is used when calculating A? Why is it not caused by controlling K?

 The solutions of nPRM are theoretically reasonable since they depict the behaviors of the dynamic system on 4 conditions separately. The old PRM solutions are not so clear, moreover, two cases (B and D) do not exist. The nPRM is better than PRM not only because it describes all the cases more clearly but also because the whole procedure applying nPRM is more sophisticated. For example, we add a smoothing step to eliminate the discrepancy between differentiable assumption and discrete observations. Also, the calculation of k is more polished than the original PRM.

We identify these solutions from the phase plane plots in Figures 1 and 2, which is commonly used when describing the behaviors of differential equations. These solutions are obtained theoretically by solving equation (1). During the solving process, the A and k are constant in the short time period t in [t0, t1]. Consequently, calculating A and controlling k are not supposed to affect the solutions in each period. However, when we dynamically model the stock prices with changing time periods, we need to consider the changes of A and k.

 We further reply to the next comment (comment 7) for more details in estimating A and k.

7. In nPRM, A and K are the most critical parameters, and their estimation method will directly determine the predicted performance. Authors should focus on how to estimate A and K. How to determine the optimal A and K is the key. From the prediction results, it is unreasonable to assume that the index obeys geometric Brownian motion.

 We think the assumption that stock index follows the geometric Brownian motion is reasonable. This is why that in the previous edition of manuscript we simulate sample paths of a stock index prices from the geometric Brownian in our Simulation Design Section (line 260-262 in the previous edition).

We re-assess the estimations of the coefficients, M (smoothing period) and A (equilibrium price), in this revision. In this revision, we determine the coefficients in the training dataset of 80% historical daily closing prices. As a result, we revise the Result Section and delete the Simulation design subsection.

In details:

Results (line 250-254):

“For application to real data, we split the data into a training set and a testing set, consisting of the earlier 80% and the latest 20% of the data, respectively. We use the training set to specify adequate smoothing period and the time interval for estimation of the equilibrium A, that is, the appropriate choices of M and N in (7) and (8), respectively.”

Applied data (line 263-272):

“We implemented the nPRM with a few choices of M and N to obtain τ-step forecasts for the training dataset of each stock index. The values of M range from 2 to 10 while the choice of N includes 30, 60, 90, 120, and 180. We set the number of forecasting steps τ = 5. We assess all combinations of M and N by the criterion of the grand mean of MAPEs across 1-step to 5-step forecasts in the training set. Afterwards, we select the combination of the best performance as the coefficient setting for prediction in the testing set. Table 2 shows that the best combination of M and N with the smallest grand mean of MAPEs in the training set for all the ten indexes. In the following, the forecasts are conducted with M and N specified for each stock index in Table 2.”

8. In the results discussion, authors say that if applying the nPRM to the markets that are less efficient, the model coefficients may acquire more information with economic meanings and obtain more accurate forecasts than the martingale. What is the reason for this explanation? What is the definition of market efficiency? How is the author identified? If nPRM performs better in an inefficient market, why don't the authors consider using other indexes? If nPRM can perform better in other samples, I will think that this method is meaningful and contributing.

 We do not focus on the market efficiency in this work. The main contribution of this work is to present a model to characterize the stock movement with economically meaningful coefficients.

 This discussion (in line 325-327, 

“Despite the absence of the test in this study, the low errors of the martingale for the four Asian stock indexes imply that the markets tend to be efficient during the period of the data.”)

is a conjecture based on reference [2], which we do not present more tests (the generalized spectral martingale (EV-MDS) test or the wild bootstrap Automatic Variance Ratio (AVR) test in reference [2], for example) and empirical results in this article. 

The statement (in line 331-333, 

“If applying the nPRM to the markets that are less efficient, the model coefficients may acquire more information with economic meanings and obtain more accurate forecasts than the martingale.”)

could be a future work if we further research into the performance of the nPRM in markets with different levels of efficiency.

In the Introduction Section (line 17-20), we summarize research results from reference [2, 4, 5] and state

“The martingale depicts the stock movements in the strong efficient markets by the theory of random walks [4]. In contrast, many real markets are considered as the weak or the semi-strong efficient markets [2, 5]. Consequently, to forecast the prices in these markets is possible.” 

Additionally, the reference [2] concludes in the article that “The EV-MDS test result shows that all the 14 return series under study are not MDS, suggesting the potential of predictability using either linear or non-linear function of past values. … Although all markets are reported earlier to be non-MDS, the AVR test shows the return series of Bangladesh, China, Hong Kong, Malaysia and Taiwan are white noise.” 

 We add some more examples of Asian stock indexes in this revision including the Philippines PSEi Index, Thailand SET Index, India S&P BSE SENSEX, Singapore Straits Times Index (STI), Indonesia JKSE, and Malaysia KLCI. No matter how efficient an Asian stock market in our investigation is, our proposed method nPRM can predict the stock prices with relatively low MAPE.

In details:

Applied data (line 256-260):

“The empirical study was conducted based on information from real daily closing price data of ten stock indexes from 2009 to 2019. The indexes include Taiwan TAIEX, Japan Nikkei 225, Korea KOSPI, Hong Kong Hang Seng Index, the Philippines PSEi Index, Thailand SET Index, India S&P BSE SENSEX, Singapore Straits Times Index (STI), Indonesia JKSE, and Malaysia KLCI.”

Empirical results (line 283-313)

“Figure 3 displays the 1-step forecasts of the nPRM for Hong Kong Hang Seng Index along with the market values. Figure 3 suggests that the nPRM possesses accurate forecasting ability. The patterns of the forecasts from 2-step to 5-step by the nPRM are all similar to Figure 3, and thus we omit the demonstration here. Figures 4-12 illustrate the 1-step forecasts of the nPRM for other Asian stock indexes individually, which also indicate accurate forecasts by the nPRM.”

…

“Fig 7.- Fig 12.”

“Table 3 displays the MAPEs of applying the investigated methods to all of the ten Asian Indexes from 1-step forecasts to 5-step forecasts. Since the performances of three deviation-type errors (MAPE, MAE, and RMSE) are similar, we simply report and discuss the results of the MAPEs. From Table 3, the martingale (see details in S1 Appendix) has the smallest errors followed by those from the nPRM, and the curve fitting (see details in S2 Appendix) has the worst performance.”

Empirical results (line 325-331)

“Despite the absence of the test in this study, the low errors of the martingale for these ten Asian stock indexes imply that the markets tend to be efficient during the period of the data. The martingale suggests that no information we can obtain from the past to forecast the future stock prices. However, our proposed method nPRM still captures the trend of the stock movements quite well, and provides meaningful coefficients, A and k, as well under the circumstances.”

---

## [Decision Letter · Decision Letter 2]

7 Feb 2022

PONE-D-21-10180R2Study of Asian indexes by a newly derived dynamic modelPLOS ONE

Dear Dr. Lee,

Thank you for submitting your manuscript to PLOS ONE. After careful consideration, we feel that it has merit but does not fully meet PLOS ONE’s publication criteria as it currently stands. Therefore, we invite you to submit a revised version of the manuscript that addresses the points raised during the review process.

We look forward to receiving your revised manuscript.

Kind regards,

Junhuan Zhang, PhD

Academic Editor

PLOS ONE

Reviewers' comments:

Reviewer's Responses to Questions

**Comments to the Author**

1. If the authors have adequately addressed your comments raised in a previous round of review and you feel that this manuscript is now acceptable for publication, you may indicate that here to bypass the “Comments to the Author” section, enter your conflict of interest statement in the “Confidential to Editor” section, and submit your "Accept" recommendation.

Reviewer #4: (No Response)

2. Is the manuscript technically sound, and do the data support the conclusions?

Reviewer #4: Partly

3. Has the statistical analysis been performed appropriately and rigorously? 

Reviewer #4: Yes

4. Have the authors made all data underlying the findings in their manuscript fully available?

Reviewer #4: Yes

5. Is the manuscript presented in an intelligible fashion and written in standard English?

Reviewer #4: Yes

6. Review Comments to the Author

Reviewer #4: The authors made corresponding revisions to the article according to the revision comments, but some comments were not resolved.

(1) At the end of the introduction, the authors list two contributions. But they are not concrete contributions from the point of view of expression. Contributions should include the innovations of this paper, please refer to expressions in other literature.

(2) In the model section, the authors still haven't told me what the economic intuition of nPRM is. The authors only responded with an explanation of the variables in the model, but did not point out that it is reasonable for prices to change according to nPRM or is in line with economic meanings.

(3) In the results section, the authors selected ten stock index samples to test the explanatory power of the model. Although the ten stock markets were all market efficient during this period, the degree of market efficiency certainly varied. Therefore, the authors can compare the prediction accuracy of nPRM in terms of the degree of market efficiency. Empirical research needs a good result. If your model's prediction performance is not as good as existing methods, then you need to improve the model or limit the scope of use of the model.

(4) The authors proposed a newly PRM, but the comparison with the PRM was removed from the comparison of the prediction results. What is the reason? Is the comparison unsatisfactory or impossible to compare? I need a detailed comparison process.

7. PLOS authors have the option to publish the peer review history of their article (what does this mean?). If published, this will include your full peer review and any attached files.

Reviewer #4: No

---

## [Author Response · Author response to Decision Letter 2]

22 Feb 2022

Submission No: PONE-D-21-10180R2

Dear Editor,

 We greatly appreciate the reviewers’ time and effort. Your valuable comments and suggestions help us improve our paper. After careful consideration of these comments, we have addressed those comments one by one in the following responses. The original comments from the reviewer(s) are in Black and our responses are in Blue.

With Regards,

Yong Shiuan, Lee

Response to Comments from Reviewer #4

1. At the end of the introduction, the authors list two contributions. But they are not concrete contributions from the point of view of expression. Contributions should include the innovations of this paper, please refer to expressions in other literature.

 We appreciate the reviewer’s comment. In the nPRM, the solutions of the equation were derived rigorously so that the solutions of this model can better fit the realistic stock prices of the markets than the PRM (please refer to line 135-137). A procedure is developed to process the raw stock prices without loss of low frequency of data. In summary, under the same dynamic model we proposed before, the nPRM seeks for the solutions that will not only achieve higher prediction accuracy but also provide useful economic meanings for possible applications in general practice. In addition, the nPRM can make the predictions using only raw stock prices without the need to transform them into usual economic variables such as returns or price increments. This is one of the reasons why we compared our results with two commonly used methods dealing with raw stock prices, the curve fitting and the martingale. To emphasize the contributions of this work, we revised the introduction section (please refer to line 87-98).

In detail:

Introduction (line 87-98):

“For the same dynamic model described using the differential equation in the PRM, we derive its solutions rigorously so that the solutions of the nPRM can better fit the realistic stock prices of the markets than the PRM. Two major contributions of the nPRM are as follows. Firstly, the nPRM seeks for the solutions that will not only achieve higher prediction accuracy but also provide useful economic meanings (refer to Figures 1 and 2 below). The coefficients in the nPRM, A and κ, represent the implied equilibrium price, and the convergence speed to the implied equilibrium price, respectively. Secondly, for many predictive models involved in the time series analysis, it is necessary to transform the stock prices to other economic variables, usually the returns or the price increments. The nPRM can make the predictions directly for the stock prices, which implies that the model considers all daily information in the analysis. Furthermore, the conduction of the proposed procedure is simple and convenient.”

2. In the model section, the authors still haven't told me what the economic intuition of nPRM is. The authors only responded with an explanation of the variables in the model, but did not point out that it is reasonable for prices to change according to nPRM or is in line with economic meanings.

We are grateful for the reviewer’s comment. The economic intuition is indeed important for the readers to understand the nPRM. Here, we tried our best to properly address the economic intuition of the nPRM by the phase plots shown in Figures 1 and 2. These two figures are used to demonstrate that it is reasonable for the prices to change by the nPRM for a high degree of consensus among most of the market participants. The economic meanings for the four cases of the nPRM can be explained as below. For case 1 (the 1st zone in Figure 1), most market participants agree that the current price could be overestimated. Therefore, most participants expect that the future price would decrease dramatically for case 1. For case 2 (the 2nd zone in Figure 1), most participants agree that the current price could be underestimated and hence expect a sharp increase in the future price. On the condition (in both case 1 and case 2), participants have high level of consensus for the price. However, the expected equilibrium price is still unknown. The coefficient κ is the gradient value, which represents the blow-up speed of the differential equation. It is positive for both cases 1 and 2, which indicates that the future change of the price will be fierce.

For cases 3 and 4 (the 3rd and 4th zone in Figure 2), most participants think that A is the reasonable expected price. Hence, future price will converge to A. This also explains why we state that the expected price A is the implied equilibrium price in the nPRM. The gradient value, κ, is negative, and its absolute value represents the speed of convergence to the expected price A. The existence of the implied equilibrium price also means that the market is highly efficient. The empirical study shows that the prediction error (MAPE) of the nPRM can be close to the MAPEs of the martingale, and the difference between them is less than 1%.

3. In the results section, the authors selected ten stock index samples to test the explanatory power of the model. Although the ten stock markets were all market efficient during this period, the degree of market efficiency certainly varied. Therefore, the authors can compare the prediction accuracy of nPRM in terms of the degree of market efficiency. Empirical research needs a good result. If your model's prediction performance is not as good as existing methods, then you need to improve the model or limit the scope of use of the model.

 Although we do not test the market efficiency in this article (eg. the generalized spectral martingale (EV-MDS) test in reference [2]), we try to provide a glimpse of the market efficiency from sample entropy, a measure of disorder. Sample entropy quantifies the amount of regularity and the unpredictability of fluctuations in a time series. Hence, a low value of sample entropy indicates that the series is deterministic while a high value reflects randomness. We provide an extreme example in our revised manuscript to show a possible range of sample entropy. The example is a simulated series of length 180 from the standard normal distribution, namely N(0, 1). By repeating 1000 times, we obtain the average sample entropy as 2.21. On the other hand, the value of sample entropy is 0, which corresponds to a perfectly regular time series.

 We add Table 5 to present the mean value of sample entropy calculated from all series used in estimation of A and κ to make forecasts dynamically in the testing set. Basically, all the mean values are close (nearly between 0.6 and 0.8), which indicates that these ten markets exhibit market efficiency of similar levels. 

From the order of the mean sample entropy values, the prediction accuracy of the nPRM mostly agrees with the degree of market efficiency excep the Taiwan stock index TAIEX. TAIEX has a mean sample entropy value of 0.7052 (the second highest). The value suggests that the fluctuations of TAIEX is more unpredictable. However, the nPRM performs well in predicting TAIEX with MAPEs as 0.76, 1.00, 1.21, 1.43, 1.65 for 1-step to 5-step forecasts respectively. This means that the changes of TAIEX are still consistent with the movement that the nPRM characterizes.

The differences between the martingale and the nPRM are close, and the difference between them is less than 1%. Additionally, the coefficients in the nPRM has their own economic meanings. We will integrate the nPRM with other models (machine learning models, for example) to see if we can reduce the prediction error in the future.

In detail:

Results – Empirical results (line 325-353):

“Apart from the martingale, we can investigate the level of market efficiency by sample entropy [38, 39]. Sample entropy measures complexity. When we have a time series of length n as {S(1), S(2),…, S(n)} , a template vector of length m is defined as

, a template vector of length m is defined as Sm(i) = {S(i), S(i + 1) ,…,S(i+m-1)} and the distance function d[Sm(i), Sm(j)], i ≠ j is defined as the Chebyshev distance. Then, we define the sample entropy of the series, SampeEn, as

SampEn = − log\\frac{Bm+1}{Bm},

where Bm is number of template vector pairs having d[Sm(i), Sm(j)]≤r, and r is the tolerance.

The value of sample entropy is nonnegative. The value 0 of sample entropy indicates that the series is perfectly regular without noise while a high value reflects randomness and unpredictability. If we examine the market efficiency from the perspective of sample entropy, a high value of sample entropy indicates a market with high level of efficiency. To show how large the possible value of sample entropy from a highly random series can be, we simulate a series of length 180 from a normal distribution with zero mean and unit variance for 1000 times. The average value of sample entropy is 2.21.”

“Table 5 shows the mean value of sample entropy calculated from all series used in estimation of A and κ to make forecasts dynamically in the testing set for ten indexes in our empirical study. The mean values range between 0.5952 (Korea KOSPI) and 0.7833 (the Philippines PSEi), which represent similar complexity among these markets. We suspect that the forecasting errors should be larger for markets with higher level of efficiency since it is harder to predict the series with random noise. By the order of the mean sample entropy values and the MAPEs for the ten stock indexes, the nPRM shows a tendency toward better forecasts for series with lower market efficiency. That is, the forecasting ability of the nPRM usually increases as the mean sample entropy decreases. However, there is an exception of Taiwan TAIEX. The mean sample entropy obtained from the TAIEX series is the second highest, so that we won't expect a smaller prediction error among the ten indexes. However, the MAPEs of the nPRM for 1-step to 5-step forecasts are relatively low. The changes of TAIEX are similar to the movement which is characterized by the nPRM among the ten markets.”

4. The authors proposed a newly PRM, but the comparison with the PRM was removed from the comparison of the prediction results. What is the reason? Is the comparison unsatisfactory or impossible to compare? I need a detailed comparison process.

 The details of comparison between the original PRM and the nPRM are listed below. We present the MAPE (%) of the original PRM and the nPRM for the testing set in the table, where the numbers of the nPRM are exactly the same as those in Table 3 in our manuscript (page 10).

 τ = 1 τ = 2 τ = 3 τ = 4 τ = 5

Nikkei 225 nPRM 1.04 1.27 1.53 1.78 2.05

 PRM 0.98 1.57 2.16 2.82 3.58

Hang Seng nPRM 1.12 1.44 1.77 2.11 2.43

 PRM 1.09 1.68 2.21 2.76 3.22

TAIEX nPRM 0.76 1.00 1.21 1.43 1.65

 PRM 0.79 1.28 1.75 2.20 2.68

KOPSI nPRM 0.86 1.10 1.38 1.62 1.84

 PRM 0.81 1.26 1.70 2.15 2.63

PSEi nPRM 1.46 1.71 1.93 2.16 2.35

 PRM 1.55 2.17 2.63 3.19 3.70

SET nPRM 1.18 1.47 1.70 1.98 2.20

 PRM 1.16 1.59 1.98 2.35 2.58

BSE SENSEX nPRM 1.31 1.62 1.88 2.20 2.42

 PRM 1.32 1.88 2.34 2.84 3.28

STI nPRM 1.19 1.52 1.77 2.09 2.32

 PRM 1.36 1.81 2.20 2.67 2.99

JKSE nPRM 1.27 1.42 1.57 1.73 1.84

 PRM 1.27 1.73 2.14 2.49 2.80

KLCI nPRM 0.98 1.30 1.58 1.89 2.07

 PRM 1.07 1.50 1.93 2.44 2.80

This table shows a similar phenomenon that the PRM has a tendency toward higher prediction error for multiple step forecasts. Although some (4 out of 10) of the MAPEs of 1-step forecast by the PRM are smaller than those by the nPRM, it simply happens by chance when the cumulated error is still small. The results of the above Table are not significant on the conclusions of this study.

 The comparison has been made in the manuscript of our previous version. Here, we quote some sentences as follows.

“Comparing the nPRM with the original PRM, although the PRM obtains smaller 1-step forecast errors, the PRM obtains larger errors than the nPRM with the increase of the number of forecasting steps.”

“The original PRM has some flaw in the integration process and does not require smoothing for the stock price. These disadvantages cause errors in forecasts. The errors are small when the time interval of forecast is short, but they grow with the increase of the time length. The newly proposed method nPRM improves these weaknesses. Therefore, the forecasting errors are notably smaller than the PRM for multiple step forecasts.”

We removed this comparison due to limited space. It’s fair to compare the proposed method with other commonly used models instead of a method we proposed previously, which is the more general way to demonstrate the empirical study.

---

## [Decision Letter · Decision Letter 3]

24 Mar 2022

Study of Asian indexes by a newly derived dynamic model

PONE-D-21-10180R3

Dear Dr. Lee,

We’re pleased to inform you that your manuscript has been judged scientifically suitable for publication and will be formally accepted for publication once it meets all outstanding technical requirements.

Kind regards,

Junhuan Zhang, PhD

Academic Editor

PLOS ONE

Additional Editor Comments (optional):

Reviewers' comments:

Reviewer's Responses to Questions

**Comments to the Author**

1. If the authors have adequately addressed your comments raised in a previous round of review and you feel that this manuscript is now acceptable for publication, you may indicate that here to bypass the “Comments to the Author” section, enter your conflict of interest statement in the “Confidential to Editor” section, and submit your "Accept" recommendation.

Reviewer #4: All comments have been addressed

2. Is the manuscript technically sound, and do the data support the conclusions?

Reviewer #4: Yes

3. Has the statistical analysis been performed appropriately and rigorously? 

Reviewer #4: Yes

4. Have the authors made all data underlying the findings in their manuscript fully available?

Reviewer #4: Yes

5. Is the manuscript presented in an intelligible fashion and written in standard English?

Reviewer #4: Yes

6. Review Comments to the Author

Reviewer #4: The authors have made improvements to all the issues mentioned in the revision comments. The article basically meets the publication standards.

Minors：

The tense of verbs: This proposed procedure brought (abstract), “PRM tend (line 84)”.

7. PLOS authors have the option to publish the peer review history of their article (what does this mean?). If published, this will include your full peer review and any attached files.

Reviewer #4: No

---

## [Editor Report · Acceptance letter]

21 Apr 2022

PONE-D-21-10180R3 

Study of Asian Indexes by a Newly Derived Dynamic Model 

Dear Dr. Lee:

I'm pleased to inform you that your manuscript has been deemed suitable for publication in PLOS ONE. Congratulations! Your manuscript is now with our production department. 

Kind regards, 

on behalf of

Dr. Junhuan Zhang 

Academic Editor

PLOS ONE